

**Calcite and vaterite biosynthesis by nitrate dissimilating bacteria in**
**carbonatogenesis process under aerobic and anaerobic conditions**
Marwa Eltarahony[1], Sahar Zaki[1]*, Ayman Kamal[1], Desouky Abd-El-Haleem[1]
[1]Environmental Biotechnology Department, Genetic Engineering and Biotechnology Research Institute (GEBRI),
City of Scientific Research and Technological Applications (SRTA-City), 21934, New Borg El-Arab City,
Alexandria, Egypt
*Correspondence to*: Sahar A. Zaki (saharzaki@yahoo.com)





**Abstract**
This study deals with 16S rDNA identified bacteria, *Lysinibacillus sphaericus* (71A), *Raoultella planticola*
(VIP), and *Streptomyces pluricolorescens* (EM4) capable of precipitating $CaCO_3$ through a nitrate reduction
aerobically and anaerobically. The produced $CaCO_3$ crystals were analyzed using XRD, EDX, and SEM. The
results showed that the carbonatogenic bacteria served as nucleation sites for $CaCO_3$ precipitation with distinct
variation in polymorph and morphology; reflecting strain-specific property. Notably, the amount of precipitated
$CaCO_3$ recorded 3.27 (aerobic), 1.55 (anaerobic), 4.15 (aerobic), 3.75 (aerobic) and 1.87 (anaerobic) g/100 mL of
strains 71A, EM4 and VIP, respectively, for 240h of incubation. The study of changes in media chemistry during
carbonatogenesis process revealed positive correlation between bacterial growth, nitrate reductase activity, pH,
EC, amount of deposited $CaCO_3$ and $NO_3^-$ consumption. Therefore, the applications of these bacterial strains,
which employed for the first time in carbonatogenesis process, are promising in the environmental, biomedical
and civil engineering fields.
**Key words:** *Streptomysetes*, $CaCO_3$ biodeposition, carbonatogensis process, nitrate reduction,
biocementation.













## 1. Introduction


Biomineralization is a process of inorganic mineral deposition by living organisms, which occurs
naturally at a slow rate over geological times. Microorganisms mediate the biomineralization process
through a sequence of biochemical activities and physiological pathways, which alter the chemical
environment and ultimately lead to mineral precipitation (**Chaparro-Acuña et al., 2018**), by two main
different mechanisms; biologically controlled mineralization (BCM) and biologically induced
mineralization (BIM) **(Ghosh et al., 2019, Wei et al., 2015; Anbu, et al., 2016**).
Interestingly, Microbial induced calcium carbonate precipitation (MICCP) which also called
carbonatogenesis attracted a considerable attention in various biotechnological applications.
Carbonatogenesis is an eco-friendly and cost-effective technology that can be applied to remediate
various environmental pollution originated from anthropogenic activities (**Rodriguez-Navarro et al.,**
**2012)**. As referred by **Chaparro-Acuña et al., (2018).** It enhances water quality in water softening
process, which subsequently participates in solving water crisis problem. On industrial level, calcium
carbonates have been widely used as viscosity modifier in plastics, rubber, inks, paint, paper and
pigment products (**Anbu et al., 2016**). For medical and therapeutic sectors, it has been utilized in drug
delivery and tissue engineering **(Poelvoorde, 2017)**. Recently, microbial $CaCO_3$ paved the way for
new subdiscipline in biotechnology, which is construction microbial biotechnology, including
biocrusting, and biocementation **(O'Donnell et al., 2019**).
Naturally, calcium carbonate occurs on earth's surface, and contributes mainly in geochemical reservoir
for carbon (**Hu et al., 2012**). It exists in various polymorphs with distinct characteristics, including
vaterite (the most thermodynamically unstable and the highest solubility spherical like), aragonite (the
densest and thermodynamically unstable needle like), calcite (the most stable rhombic), two hydrated
crystalline phases, monohydrocalcite, ikaite and amorphous phases (**Sevcık et al., 2018**). The
metastable phases can be easily recrystallized to stable calcite phase (**Han et al., 2017**). As reported in
extensive studies (**Ersan et al., 2015**, **Ghosh et al., 2019**), different microorganisms precipitated
different types of $CaCO_3$.





The calcifying microorganisms stimulate $CaCO_3$ precipitation via two fundamental mechanisms either
autotrophic or heterotrophic (**Singh, 2019**); which seems to be more abundant. Photosynthetic
microorganisms fix $CO_2$ and induce carbonate precipitation autotrophically (**Richardson et al., 2014**).
Conversely, three main categories of microorganisms induce biocalcification process heterotrophically.
The first category catalyzes the reduction of sulphate by sulphate reducing bacteria (SRB) (**Lin et al.,**
**2018).** The second category comprises microorganisms, which participate in nitrogen cycle by one of
the following means: A)- oxidative deamination of amino acids, B)- nitrate reduction C)- urea
hydrolysis (**Richardson et al., 2014**). The third category promotes the reversible conversion of $CO_2$ to
bicarbonate through carbonic anhydrase enzyme (**Zhu and Dittrich 2016**).
Remarkably, the majority of studies concerned with calcification technology focused on ureolysis
processes and few researches were performed on nitrate dissimilation metabolism (**Zhu and Dittrich**
**2016).** Nonetheless, ureolysis processes exhibited several limitations, namely; the byproduct of urea
hydrolysis (ammonia or ammonium), which is potentially hazardous, requires removal later on by
another stage (**O'Donnell et al., 2019**). Moreover, the using of aerobic urolytic bacteria *in situ* will
result in calcite disintegration due to oxygen shortage and changing in pH surrounding to bacteria,
which eventually lead to insufficient applications (**Thirumalai, 2015**). Interestingly, carbonatogenesis
via dissimilatory nitrate reduction deemed as remarkable alternative mechanism that can overcome the
drawbacks of ureolysis. Where, nitrate dissimilatory microorganisms are more prevalent in the
subsurface and display flexibility in their growth strategy; they are able to utilize low $NO_3^-$
concentrations under anoxic conditions and without formation of harmful or toxic byproducts. Besides,
denitrification is thermodynamically more favorable than ureolysis. As refereed by **Ersan et al., 2015,**
the change in standard Gibbs energy for denitrification is -785 kJ/mol acetate, while it was estimated to
be -27 kJ/mol acetate for ureolysis. Furthermore, the carbonate yield generated by denitrification
process is higher than ureolysis**.**
Accordingly, the present study aimed to determine the carbonate precipitation efficiency of
heterotrophic nitrate dissimilating bacteria under both aerobic and anaerobic conditions. The selected





nitrate reducing-bacteria under study were isolated from Egyptian non-calcareous habitats and identified
by 16S rDNA gene sequencing. The substantial part of this study focused on characterization of CaCO₃
precipitated by each strain under oxic and anoxic conditions, which will check the suitability of each
CaCO3 crystals considering their prospective application according to mineralogy and morphology.
Subsequently, different criteria such as bacterial count, nitrate reductase (NR) activity, pH, deposited
CaCO₃ amount, $NO_3^-$ concentration, $NO_2^-$ concentration and electrical conductivity (EC) were analyzed.
As far as the authors know, it is the first report of carbonatogenesis process through nitrate reduction
under aerobic and anaerobic conditions.

## 2. Materials and Methods

### 2.1. Sampling, screening and selection of nitrate-reducing bacteria

Sediment samples were collected from non-calcareous Egyptian sites; Naba Alhamra at Wadi Elnatron
(Al-Beheira governorate), Karon Lake (Al-Fayoum governorate) and Mariot Lake (Alexandria
governorate). Directly after sampling, isolation and screening of bacteria for nitrate reduction were
performed. Initially, 1 g of fine powdered homogenized samples were serially diluted in 0.85 % saline,
and then plated on denitrifying media containing bromothymol blue and incubated aerobically (**Lv et**
**al., 2017**). The bacterial isolates that reduced nitrate aerobically were picked up and re-examined for
nitrate reduction under anaerobic conditions as described by **Zaki et al. (2019)**. Out of 17 nitrate
reducing-bacteria, three isolates, designated as 71A, VIP and EM4 were selected based on their nitrate
reduction capabilities. Generally, nitrate reductase (NR) assay performed using spectrophotometric
measurement of nitrite concentration at 540 nm. This was based on diazo-coupling method with
Griess reagents (0.2 % Naphthyl ethylenediamine and 2% Sulfanilamide in 5% phosphoric acid).
Nitrite generated from nitrate in presence of 40 mM of an artificial electron donor dithionite
benzyle viologen. One unit of NR activity corresponds to the amount of enzyme that catalyzes
the formation of 1μmol of nitrite per minute or 1μmol of nitrate reduced per minute under
standard assay conditions (**Zaki et al., 2019**).

### 2.2. Molecular identification of selected isolates



The selected isolates were identified using16S rDNA sequencing. The bacterial genomic DNAs of the
selected isolates were extracted from overnight pure cultures and 16S rDNA genes were PCR amplified,
purified and sequenced as described elsewhere **(Vashisht et al., 2018).** The phylogenetic affiliation was
inquired by applying BLAST analysis to determine the similarities with their available GenBank
database sequences. Their generated sequences were submitted to the GenBank to obtain corresponding
accession numbers. For multiple alignment and phylogenetic tree construction, the software package
MEGA- 6 was employed.

### 2.3. CaCO$_3$ precipitation and crystals collection

The capability of selected isolates for CaCO$_3$ precipitation through nitrate reduction process was
assessed in liquid broth method at flask level. About 250 µL of bacterial cultures (1.8 x $10^6$ CFU/mL)
were inoculated in 200 ml CaCO$_3$ precipitation media (CCP) which composed of M9 media
supplemented with (g/L): sodium acetate (10) and Ca (NO$_3$)$_2$·4H$_2$O (15) at pH 7.0 ± 2.2 **(Ersan et al.,**
**2015)**. The flasks were incubated aerobically in an orbital shaker at 150 rpm and anaerobically as
stated by **Zaki et al., (2019)**. The inoculated flasks were incubated at 30°C for 10 days. An abiotic
negative control consisted of un-inoculated media was run in parallel. At the end of the experiment, the
whole cultures were centrifuged at 10.000g for 20 min and washed successive times by distilled water
and ethanol to eliminate any nutritive solution. The air-dried minerals were weighted to estimate the
amount of precipitated CaCO$_3$ and subsequently subjected to mineralogical studies **(Vashisht et al.,**
**2018).**

### 2.4. Mineralogical and morphological analysis

The mineralogical analysis of the dried precipitated CaCO$_3$ was established with X-ray diffraction
(XRD), Energy dispersive X-ray spectroscopy (EDX) and scanning electronic microscopy (SEM). The
mineral phase of precipitated CaCO$_3$ was identified using X-ray diffractometer ((Bruker MeaSrv D2-
208219, Germany-Central Lab, Faculty of science, Alexandria University) that operating with Cu
Kα radiation (λ = 0.15406 nm) generated at 30 kV and 30 mA with scan rate of 2°/min for 2θ values




over a wide range of Bragg angles 10°≤ 2θ ≤ 80. The microchemical sample analysis was carried out
using EDX analyzer combined with SEM (JEOL JSM 6360LA, Japan). The morphological
characteristics of bacterial CaCO₃ was observed using SEM (JEOL JSM 6360LA, Japan – Advanced
Technologies and New Materials Research Institute (ATNMRI) SRTA-City) at an accelerating
voltage of 20 kV **(Silva-Castro et al., 2015).**

**2.5. Study of the parameters associated with CaCO₃ precipitation**

The correlation between CaCO₃ formation and the parametric changes in culture media during different
growth phases of all strains under study were investigated. The parameters; bacterial count, NR activity,
concentrations of $NO_3^-$, $NO_2^-$, pH, electrical conductivity (EC), and weight of precipitated CaCO₃ were
screened at constant time intervals. Strains were inoculated on the media, which were reported formerly
and incubated at 30°C both aerobically and anaerobically for 10 days. At each time interval (6 h), about
15 mL aliquot of the culture was drawn and subjected for analysis. Pour plate method was applied for
assessing the bacterial count (CFU/ mL) on nutrient agar and incubated overnight at 30°C. The
precipitated CaCO₃ was collected by centrifugation at 10.000 g for 15 min, washed with sterile distilled
water, air dried and weighed. The supernatant was used to determine the rest of parameters, where, pH
values were measured using a pH indicator (PB-10, Sartorius AG), while EC measured using electric
conductivity meter (JENWAY- 4510). The concentrations of $NO_3^-$ and $NO_2^-$, were measured according
to the procedure followed by **APHA, (1999)**.
3. **Results and Discussion:**
**3.1. Isolation and identification of bacteria**
Among 17-screened bacterial isolates, three of them 71A, VIP and EM4 were selected based on their
high NR activity. Then isolates were subjected for taxonomic identification and examination of
carbonatogenesis process. The partial 16S rDNA sequences of 1127, 1025 and 800 bp of isolates 71A,
VIP and EM4 exhibited 98.4, 97.2 and 99.8% DNA similarities with *Lysinibacillus sphaericus*,
*Raoultella planticola* and *Streptomyces pluricolorescens,* respectively. Their 16S rDNA sequences were





deposited in the GenBank under accession numbers MK936472 (71A), MK551748 (VIP) and
KY964509 (EM4). Strain 71A is belonging to the phylum Firmicutes and family Bacillaceae. Whereas,
the taxonomic affiliation of VIP and EM4 are belonging to phylum Proteobacteria, family
Enterobacteriaceae and phylum Actinobacteria, family Actinomycetaceae, respectively. As pointed out
by **Silva-Castro et al., (2015)**, members of Firmicutes phylum are the most predominant in MICCP
process through ureolysis. Besides, **Talaiekhozani et al., (2014)** referred to the calcification potency of
ureolytic *Proteus vulgaris* in concrete self-healing, which grouped in family Enterobacteriaceae.
Additionally, some genera affiliated to Actinobacteria deposited $CaCO_3$ based on metabolizing
nitrogenated organic substrates such as peptone and yeast extract **(Torres et al., 2013).** The
phylogenetic tree of the selected strains was constructed by the Neighbour-joining (NJ) method as
indicated in Fig. 1.
**3.2.  Nitrate Reductase activity (NR)**
Actually, among 17 screened isolates, *L. sphaericus* (71A), *R. planticola* (VIP), and *S. pluricolorescens*
(EM4) showed the maximum NR activity, after 24 h of incubation exhibiting 449, 534 and 768 µmole/
min/ml, respectively under aerobic conditions.  However, under anaerobic conditions and on the 1[st] day
of incubation, NR activity was 189 and 426 µmole/min/ml with strains 71A and VIP, respectively. In
general, the NR activity of both strains was increased along with the incubation time as mentioned later
on. On the other hand, strain EM4 did not show NR activity anaerobically, while it exhibited the highest
NR activity aerobically. Therefore, it was selected. Our knowledge, there were no previous studies
reported *Streptomyces* species in carbonatogenesis process through nitrate dissimilation pathway.
**3.3.  CaCO₃ biodeposition**
Despite of almost preceding literature emphasized that the successes in isolation of CCP organisms are
in particular based on the selection of sampling sites (calcareous and cementitious); the existing
investigation did not comply with this rule. Despite, all of the isolates were isolated from non-
calcareous sites; they possessed the specified mechanism, which allows $CaCO_3$ biodeposition





(Montano-Salazar et al., 2017). The selected strains that precipitated crystals in CCP medium at 30°C
under aerobic and anaerobic conditions exhibited different appearance, which includes crystal size,
texture and color Fig. 2. Conversely, clear solution without any precipitates was observed in the abiotic
uninoculated experiment (control), implying the ability of active strains to modify the chemistry of
culture media and creating the proper microenvironments favoring $CaCO_3$ precipitation. Obviously,
large beige or buff color and irregular crystals were appeared in anaerobic cultures, whereas fine white
powder was formed in aerobic cultures of strains 71A and VIP, while, strain EM4 culture showed
yellowish- brown aggregated pellets. Interestingly, the inter-species differences in crystallization
patterns, colors, textures and forms were noticed previously by **Montano-Salazar et al., (2017)**, where,
*Rhodococcus qingshengii* M101, *Arthrobacter crystalopoyetes* and *Psychrobacillus psycrodurans*
showed spherical brown, irregular yellowish and irregular white/beige aggregated crystals of $CaCO_3$,
respectively.
To study the involvement of nitrate dissimilation in $CaCO_3$ formation, the changes in the chemistry of
media was monitored. Generally, a positive correlation was observed between the amount of
precipitated $CaCO_3$ with bacterial growth that was synchronized with NR activity, pH, EC, $NO_3^-/NO_2^-$
reduction Fig. 3. The aerobic growth resulted in the higher bacterial count, NR activity and eventually
higher amount of $CaCO_3$. Such is plausible due to the availability of higher redox potential in presence
of oxygen (+818 mV), which supports rapid energy generation, higher metabolic activity and hence
higher reproduction rate **(Ilbert and Bonnefoy, 2013)**. Evidently, the cell number increased rapidly and
reached to the maximum between 90h and 120h depending on the strain, thereafter it decreased slowly
and steadily until the end of the experiment (240h). Remarkably, upon 120h of aerobic incubation,
about 4.5x $10^8$ CFU/mL of strain 71A with maximum NR activity (779 µmol/min/mL) completely
reduced $NO_3^-$ and uplifted pH from 7.01 to 8.53.  In the same extent, the aerobic culture of strain VIP
removed $NO_3^-$ completely and elevating the initial pH from 7.01 to 8.91 at 90h by the activity of 6.65 x
$10^8$ CFU/ml, which exhibiting 862 µmol/min/mL of NR activity. Interestingly, strain EM4 (6.6 x $10^7$
CFU/ ml) displayed the highest NR activity with 1292 µmol/min/mL and increasing pH to 9.51 with
complete $NO_3^-$ reduction at 102h of incubation.




In comparison, the anaerobic cultures (9.7 x 10$^6$ and 7.1 x 10$^7$ CFU/ mL) of strains 71A and VIP
eliminated NO$_3^-$ by means of 180 and 66h, respectively. However, NR activity and pH were recorded
372 and 661 μmol/min/mL and 8.8 and 9.7 for 71A and VIP, respectively. In addition, a complete
denitrification process was achieved upon continued anaerobic incubation. Obviously, NR activity was
expressed aerobically even after the complete depletion of NO$_3^-$ and in the presence of NO$_2^-$, whereas,
under anaerobic conditions, it induced only in the presence of NO$_3^-$. That could be assigned to the
physiological role differences of NRs under different aeration conditions. Remarkably, membrane-
bound NR is induced under the absolute absence of oxygen and mainly involves in anaerobic nitrate
respiration, for production of the electrochemical proton gradient and generation of ATP (**Zaki et al.,**
**2019**). On the other hand, periplasmic NR is unaffected by oxygen level or C and N balance; it
maintains redox homeostasis by dissipating excess reductant during aerobic growth and scavenging
toxic concentrations of nitrate and nitrite as pointed out by **Li et al., (2012).** Thus, NR activity was
observed along with aerobic incubation process. Furthermore, the availability of more bacterial cell
number enables more nucleation sites, which ultimately precipitate more carbonate crystals
**(Rodriguez-Navarro et al.2007).** Virtually, the amount of CaCO$_3$ precipitates kept increasing
gradually during the mineralization process and recorded in g/100 mL, 3.27 and 1.55 for strain 71A
(aerobic and anaerobic), 3.75 and 1.87 for strain VIP (aerobic and anaerobic) and 4.15 for strain EM4
(aerobic only), by 240h of incubation. On the other hand, **Gomaa, (2018)** stated that *Micrococcus* sp.
induced 10.80 mg/ml of CaCO$_3$. Additionally, **Kaur et al., (2013)** documented that within three weeks
of incubation, *B. megaterium, B. subtilis, B. thuringiensis, B. cereus* and *L. fusiformis* produced 187,
178, 167, 156 and 152 mg/100 ml of CaCO$_3$ via ureolysis pathway, which make results of the current
study characteristic. Regarding the incubation time, a similar period for the CaCO$_3$ biodeposition was
observed by **Rodriguez-Navarro et al. (2007)** for *Myxococcus xanthus*.
Consistent with the bacterial count, NR activity and pH, a linear progressive increase in EC was
noticed. That could be ascribed to the elevation of medium conductivity (mS/cm/min) by the action of
charged ions such as NO$_2^-$, N$_2$O$^-$, NO$^-$, Ca$^{2+}$ and CO$_3^-$ ions generated by microbial activity on non-
conductive substrates (sodium acetate and Ca(NO$_3$)$_2$·4H$_2$O). It is worth mention that the conductimetric



method is mainly used in evaluation of ureolysis process to follow the generation of ionic products from
non-ionic substrates and consequently give insight on the microbial activity and mineralization
tendency. Besides, it was used to control the kinetics of nucleation and crystal growth of carbonate
precipitation process in the presence of exopolymer as referred by **Szcześ et al., (2018)**. Near the end of
the experiment, a slight decline or stability state was observed by the almost of examined parameters for
all bacterial strains both aerobically and anaerobically. That could be attributed either to entrance of the
cells in stationary phase, where no more increase in cell number as a result of nutrient depletion and
subsequently no more $CaCO_3$ precipitation, or fossilization of the cells within $CaCO_3$ crystals. The
latter case caused mineralization of bacterial cell wall, which subsequently inhibited nutrient exchange
with surrounding environment and eventually cell death as recorded by **Silva-Castro et al., (2015).**
### 3.4. $CaCO_3$ crystal analysis
XRD, EDX, and SEM techniques were employed to characterize the deposited $CaCO_3$ crystals. The
nature of crystals, crystallographic identity and phase purity of inorganic compounds were determined
using XRD. The characteristic signature peaks of calcite at 2θ values of 23.13, 29.50, 36.04, 39.51,
43.31, 47.51, 48.65, 56.71, 57.50, 60.81, 63.22, 64.42, and 65.57, respectively correlated with lattice
(hkl) indices of (012), (104) (110), (113), (202), (024),  (116), (211), (122), (214), (125), (300) and
(0012) were identified in strain 71A precipitated samples under both incubation conditions. On the other
hand, the XRD spectrum of strain VIP samples recorded calcite and vaterite under aerobic and
anaerobic conditions, respectively Fig. 4. In regards to examined sample of strain EM4, the relative
intensities and the reflection peak positions at 20.93, 24.81, 27.13, 32.75, 39.82, 42.66, 43.13, 49.85,
51.13, 55.75, 60.36, 62.54 and 65.38, which corresponds to crystallographic planes of (002), (100),
(101), (102), (103), (004), (110), (104), (200),(202), (105), (114) and (006), respectively, confirms the
presence of vaterite Fig. 4 (D & E). The diffraction peaks of calcite and vaterite match with those
of the standard spectrum JCPDS, No. 02-0629 and. JCPDS, No. 72-0506, respectively
**(Svenskaya et al., 2017).** Generally, the diffractograms of all examined samples appeared sharp,





clearly distinguishable and broad, which indicates the pure, ultra-fine nature, small crystallite size and
negating the possibility of mixed phases biominerals.
The EDX microanalysis of the bioprecipitated crystals ae presented in Fig. 5. The elemental profiles of
the examined samples exhibited typical characteristic elemental peaks at 0.277, 0.525 and
3.69keV with atomic percentages range of (17-20 %), (42-51%), and (32-40%), which is related to
the binding energies of carbon, oxygen and calcium, respectively. Additionally, there were other
EDX peaks could be noticed in a small percentage such as Na and Cl, which proposed to be ingredients
of culture media. This result is in agreement with **Han et al., (2018).** Obviously, vaterite samples of
anaerobic culture of strains VIP and EM4 displayed another additional phosphorus (P) peak (2.013
keV) with considerable percentage assessed by 2-3%. Apparently, its presence could suggest
being a biological origin, where, it represents essential constituent of bacterial biomolecules such as
phospholipids, nucleic acids, proteins and/or polysaccharides. The involvement of vaterite with P is
considered to be advantageous by providing stabilization and subsequently preventing transformation to
calcite form. The same result was obtained with other research groups (**Ghosh et al., 2019**).
Generally, the calcium peaks intensities and their corresponding atomic percentages, which were higher
than carbon peak, may reflect higher purity in structure as implied by **Caicedo-Pineda et al., (2018).**
The detailed characterization about the morphology, texture, surface and size of bio-deposited $CaCO_3$
crystals were studied by SEM. As shown in Figure 6A, approximately square or cubic shape $CaCO_3$
crystals in the range of 0.2 to 3.7 μm was noticed with strain 71A under aerobic condition. Close up
view of these crystals depicted smooth surface embedded with rod shaped bacterial cells Fig. 6 (B) and
some wrinkled surface globules with internal small holes Fig. 6 (B), indicated by arrows). Higher
magnification in another sector displayed casts of bacterial cells and rhombohedral particles cemented
in mucous matrix as referred by head arrow Fig. (6C), such mucilaginous like material could be
considered as a polysaccharide excreted by carbonatogenic bacteria. This result concurred with the
previous report in which rhombohedral calcite was produced by *B. megaterium* and embedded in slimy
matrix (**Kaur et al., 2013**). On the other hand, the calcite formed anaerobically appeared as aggregated



grains in size of 6.2 to 22.4 μm and irregular shaped clusters Fig. 6 (D). Additionally, subhedral
rhombohedral particles with defined faces and edges were observed accompanying with anhedral
crystals Fig. 6 (E). The presence of mucoid substance that encompassed these particles were also
detected Fig. 6 (E), head arrows. Further, round shaped calcified bacterial cells were evident on the
surface of bioliths Fig. 6 (F). Such change in cell morphology at anaerobic conditions could be ascribed
to unfavorable conditions that lead to sporulation of vegetative cells. Virtually, the sporulation
capability of strain 71A seemed to be advantageous; particularly in prospective applications with harsh
conditions as self-healing of concrete cracks. In fact, bacterial spores are able to withstand adverse
environmental conditions to maintain cell viability (**Vashisht et al., 2018).** However, the calcite
mineralized by aerobic culture of strain VIP exhibited coarse, imbricated subhedral, rhombohedral
minerals with size ranged from 0.79 μm to 1.63 μm Fig 6 (G). Viewed at higher magnification Fig 6
(H), the calcite crystals were accumulated compactly and assembled into stacks like structures. The
bacterial cell contours were evident on calcite surface (indicated by arrows) Fig. 6 (I). Interestingly,
calcite crystals with size range (1–10 μm) were produced during nitrate assimilation process by
saprophytic fungus *Alternaria sp.* **(Hou et al., 2011),** which is consistent with results of the present
study.
Remarkably, the vaterite of anaerobic strain VIP culture showed series of globules, spherulite crystals
with size range of 12.3 to 61.8 μm Fig. 6 (J & K). In addition, the bacterial imprints were indicated by
cavities on smooth surface of the sphere Fig. 6 (L). Such imprints emphasized the intrinsic role of
bacterial cell as nucleation site for CaCO3 precipitation, which totally concurred with the finding of **Li**
**et al., (2012)**. In coincident with our results, **Rodriguez-Navarro et al., (2012)** demonstrated the ability
of *Myxococcus* sp. to induce different morphologies of both vaterite and calcite depending on growth
conditions and medium composition. On the other hand, the spiny vaterite beads (5.5 – 77.6 μm), which
spiked with triangular sharp point surface, were formed by strain EM4 Fig. 6 (M). The magnified field
of vaterite pellets illustrated ramified CaCO3 crystals encapsulated the hyphae of *Streptomyces* cells
Fig. 6 (N & O). Similar results were obtained with **Caicedo- Pineda et al., (2018)**. Otherwise,



elongated plate-like crystals were produced by the actinomycete culture of *Thermomonospora* sp.
**(Rautaray et al., 2004).**
Notably, the bacterial cells and their corresponding metabolic activity were prerequisite for the
bioprecipitation of $CaCO_3$ crystals, particularly with absence of such deposition in abiotic negative
control **(Han et al., 2017)**. Generally, the nitrate bioreduction is the predominant mechanism in $CaCO_3$
precipitations, which was summarized in the equation (1) to be followed by the strains under study
**(Ivanov et al., 2015):**
$5\ CH_3COONa + 4\ Ca\ (NO_3)_2 \rightarrow 4\ CaCo_3\downarrow + 4\ N_2\uparrow + 6\ CO_2\uparrow + 5\ H_2O + 5\ OH^-$   Eq. (1)
In fact, the studied bacterial species reduced nitrate by NR enzymes to oxidize the organic carbon and
electron donor (acetate) for energy generation and cells proliferation. As referred by **Singh et al.,**
**(2015)** and **Zhu and Dittrich, (2016),** protons were consumed continuously during such process and
resulted in production of respired $CO_2$ and bicarbonate which ultimately elevated pH and alkalinity of
ambient medium **(Hou et al., 2011; Zhu and Dittrich, 2016).** Under this circumstance, the
precipitation/crystallization process is initiated in two main steps based on the crystal growth theory
**(Trushina et al., 2014).** The first is crystal nucleation, which a new solid phase in nanometer size forms
in supersaturated solution **(Wu et al., 2017)**. The second step is crystal growth, which could be
described as atom-by-atom addition to the newly formed nuclei. Consequently, the growth of larger
crystals and increasing in the particles size either occur randomly or oriented at the expense of smaller
crystals or nanoaggregates **(Zhou et al., 2010).** The particles with lower surface charge tend to
coagulate and agglomerate to each other in a crystallographically oriented manner till reach to the most
stable crystals with particular size that cause sedimentation. Thus, at this point the precipitation process
is completed **(Rodriguez-Navarro et al. 2007; Trushina et al., 2014)**. Substantially, such precipitation
process is not genetically or biologically controlled by the microorganism itself, but mediated by the
physico-chemical properties of the surrounding environment **(Caicedo-Pineda et al., 2018)**.





Noteworthy in this context is nucleation of crystals and solution supersaturation, which determine the
size and polymorphic form of precipitated crystals **(Vekilov, 2010).** Where, crystal nucleation
dominates over crystal growth at a relatively higher degree of supersaturation, which ultimately
generates smaller size with approximately identical shaped crystals. On the other hand, at low
supersaturation, the nucleation is slow and crystals grow faster than they do nucleate, resulting in
aggregates of large crystals of various sizes forms **(Rodriguez-Navarro et al. 2007; Wu et al., 2017).**
Such principle could explain the formation of small size deposits at aerobic cultures. In addition, it was
suggested that anaerobic cultures of strains 71A and VIP excreted certain polysaccharides with adhesive
nature that stickled tightly the fine particles into larger crystals. In agreement with these results,
**Shirakawa et al., (2011)** stated that under static conditions, culture of *L. sphaericus* precipitated larger
$CaCO_3$ crystals than in shaken cultures.
In the same extent, as pointed out by **Rodriguez-Blanco et al., (2017)**, the structure, morphology,
stability and crystallization pathway of $CaCO_3$ precursor to either vaterite or calcite governorates by
several factors. These include binding strength of $Ca^{2+}$ and $CO_3^{2-}$ ions within the $CaCO_3$ precursor
aggregates, solubility, and the dissolution rate of $CaCO_3$ precursor, which all are pH-dependent. It is
substantial to mention that alternation in pH values contributes in the ionic strength, which consequently
effects on solution saturation. Where, higher supersaturation occurs at higher pH, alkalinity and
carbonate ions concentrations. In addition, at higher supersaturation, the crystals with higher solubility
and lower stability form first and vice versa as described by the Ostwald's law of stages **(Rodriguez-**
**Navarro et al. 2007; Trushina et al., 2014).** That could explain vaterite formation by anaerobic culture
of strain VIP and strain EM4; where higher metabolic activity accompanying with nitrate dissimilation
process led to increase carbonate content and rapidly elevating pH values 9.58 (VIP) and 9.7 (EM4) to
the point of supersaturation with respect to vaterite. On the other hand, under relatively low
supersaturation, calcite formed by aerobic culture of strain VIP, aerobic and anaerobic cultures of strain
71A at pH 8.9, 8.5, and 8.78, respectively. As pointed out by **Rodriguez-Navarro et al., (2012)**, the
formation of vaterite and calcite were promoted at alkaline (8.5-10.5) and neutral pH (7), respectively.
On contrary, **Rautaray et al., (2004)** stated that synthesis of vaterite by *Verticillium sp.* was facilitated



at low pH conditions (5.3), whereas calcite is favorably formed at pH more than 10 (**Ramakrishna et**
**al., 2016).**
Besides pH, there are several key factors governed the type, crystal size and polymorph of the
biodeposited crystals including; bacterial type, nucleation sites abundance, calcium concentrations,
calcium precursor type, media composition and incubation conditions (**Anbu, et al., 2016; Chaparro-**
**Acuña et al., 2018**). Accordingly, the bacterial species was considered being the MICCP determinant in
the current study. Therefore, it is plausible to mention that bacterial surfaces properties could also
influence greatly on phase and morphology of $CaCO_3$ through heterogeneous nucleation. It is promoted
by coupling and binding of negatively charged functional (macro) molecules in the bacterial cell wall
and positively charged cations (e.g. $Ca^{2+}$). In general, the bacterial cell wall consists of peptidoglycans,
teichoic lipoteichoic acids, lipids and lipopolysaccharide, which provide the negative charge of cell
wall. As reported by **Anbu, et al., (2016),** Ca 2+ ions prevented from accumulation inside the cell and
adsorbed more frequently on cell envelope due to potency of the cell for ionic selectivity. Thereby, Ca
2+ ions were actively transferred out through passive diffusion by the action of an ATP-dependent
pump which is located close to outside of the cell. With continuous H+ uptake, higher Ca2+
concentration and higher pH are emerged surrounding the cell, creating nanoscale neighborhood that
facilitate precipitation and crystal growth of CaCO3 as described in equations 2, 3 and 4 (Li et al., 2011;
Singh, 2019):
$Ca^{+2} + cell^- \rightarrow Cell\text{-} Ca^{+2}$     Eq. (2)
$HCO_3^- \leftrightarrow H^+ + CO_3^{-2}$     Eq. (3)
$Cell\text{-} Ca^{+2} + CO_3^{-2} \rightarrow Cell\text{-}CaCO_3\downarrow$   Eq. (4)
Remarkably, heterogeneous nucleation is commonly occurred process in nature (**González-Muñoz et**
**al., 2014**). Furthermore, other bacterial components such as lipids, glycoproteins, proteins,
proteoglycans and extracellular polymeric substances (EPS) could provide additional nucleation site for
$CaCO_3$ precipitation during carbonatogenesis process (**Li et al., 2011; Ghosh et al., 2019**). Such





organic macromolecules are acidic polyanionic polymers which include carboxylic (R-COO⁻),
phosphatic (R-PO$_4^{2-}$) or sulfonate (R-SO$_3^-$) functional groups, and can serve as promoters or inhibitors
for crystals biomineralization **(Szcześ et al., 2018).**
Actually, when charged functional groups of EPS exist in a random distribution, they couple to metals
in a disordered arrangement, which obstruct crystal nucleation. However, the presence of EPS
functional groups in periodic and ordered array lead to stereo chemical gathering between the organic
matrix and the newly-formed crystals and hence promotes heterogeneous nucleation **(González-Muñoz**
**et al., 2014).**
Virtually, the culture of strain VIP might exhibit different metabolic products, proteins or EPS under
different aeration conditions, which thereafter affected on the kinetics of crystallization process. Where,
the biomolecules under certain condition may exhibit great affinity to certain face of specific polymorph
and will adsorb onto these faces, causing alterations in crystals nucleation or growth stages of the
affected polymorph on the account of the others **(Trushina et al., 2014).** Notably, EDX analysis Fig. 4
(D & E) elucidated the incorporation of phosphorus peak with precipitated vaterite of both EM4 and
anaerobic culture of VIP strains. That could explain the stability of such metastable phase and inhibition
of its conversion to stable phase (aragonite or calcite). Such biologically originated phosphorus exhibits
certain affinity to Ca$^{2+}$. As it preferentially complexes with crystal nucleus and adsorbs on specific site
during crystal growth causing growth inhibition; generating metastable vaterite **(Trushina et al., 2014).**
It is noteworthy to mention that *Streptomyces* cell wall contain teichoic, which contains 1, 5-poly
(ribitol phosphate) chain along with poly (glycerol phosphate) unites linked together by phosphodiester
bonds **(Streshinskaya et al., 2003).** Besides, it also contains diamino acid, LL-diaminopimelic acid
accompanied by glycine **(Nakamura et al., 1977),** which eventually confirm vaterite stabilization.
Several literatures documented that the preservation of bio-vaterite was favored by organophosphorous
biomolecules or phosphorus-enriched medium **(Caicedo-Pineda et al., 2018).**
Interestingly, **Tourney and Ngwenya, (2009)** shed the light on the dissolved organic carbon, which
liberated from EPS and bound with Ca$^{2+}$ ions, causing lowering of CaCO$_3$ saturation which




consequently enhances calcite precipitation over vaterite. Additionally, **Kawaguchi and Decho, (2002)**
declared that the association of specific proteins with EPS of *Schizothrix sp.* favored aragonite and
calcite polymorph selection. In the same sense, extracellular proteins excreted by the *Verticillium sp.*
and *Thermomonospora sp.* influenced significantly on both crystal morphology and polymorph
selectivity as refereed by **Rautaray et al., (2004**).
However, it was recorded that the appropriate nutrient types (e.g. carbon/ energy source and nitrogen
source) and their concentrations stimulate the bacterial growth rate and enzymatic system, and thus
provide the required chemical species and appropriate conditions for precipitation **(Kaur et al., 2013).**
Alternatively, the different calcium sources induce different mineral shape and polymorph **(Anbu, et**
**al., 2016; Kim et al., 2016; Chaparro-Acuña et al., 2018).** Where, rhombohedral calcite and disk-
shaped vaterite were induced by calcium chloride and calcium acetate, respectively. While, spherical
shape vaterite was induced by calcium lactate and calcium gluconate **(Anbu, et al., 2016).**
Herein, despite acetate and calcium nitrate supported good growth and rapid metabolic activity for
examined bacteria, but these factors had no effect on polymorphic selectivity of $CaCO_3$ minerals as it
was fixed with all examined bacterial species. In correspondence with current study, **Rothenstein et al.,**
**(2012)** reported that Ca-acetate in the culture media of *Halomonas halophila* displayed no effect on the
mineralized polymorph. Apparently, the current study has obviously shown that the variation in size,
morphology and mineral phase of the biodeposited mineral is driven by strain-specific differences.
Generally, the calcite/vaterite selectivity is a complex process and controversial issue **(Rodriguez-**
**Navarro et al. 2007; González-Muñoz et al., 2014).**
The studies on the formation of spherical vaterite crystals in synthetic systems are relatively scarce
(**Rautaray et al., 2004**, **Rodriguez-Blanco et al., 2017**). Actually, different problems encountered
during synthesis, crystallization and stabilization. In particular with its instability and rapidly
transformation into more stable phases (calcite or aragonite) at room temperature, and in an aqueous
solution. Besides, the reproducibility and shape/size control of vaterite are taken in consideration. To
overcome the above-mentioned concerns, certain additives either organic or inorganic were applied.



Nitric acid and ammonia are among inorganic additives, which deemed as facilitating factors influence
on the kinetics of vaterite production **(Trushina et al., 2014).** However, polymers such as, polyacrylic
acid, poly (vinyl alcohol), polycarboxylic acid, polyvinylpyrrolidone and commercial copolymers;
including poly (4-styrenesulfonate-co-maleic acid) (PSS-co-MA), calixarene dendrimers were the most
frequently used. Moreover, different types of alcohols and some ionic surfactant were utilized as an
effective stabilizing and polymorph controlling agents **(Trushina et al., 2014).** The traces of such
compounds associated with vaterite particles could exhibit undesired impact especially in applying
vaterite in drug delivery and pharmaceutical formulations. Thus, all the sights directed to use
biomimetically synthesized substances such as gelatin **(Wu et al., 2017),** Chitosan **(Wu et al., 2011),**
amino acids and proteins **(Trushina et al., 2014).**
Nonetheless, the natural and biological matter is even better as documented by **Wu et al., (2017).**
Where, the carbonatogenic bacteria of the current study serve as a source of carbonate ions makes this a
truly biogenic approach for minerals synthesis; hence does not consider merely biomimetic process.
Actually, the carbon dioxide utilized in biomineralization generated from metabolism of carbonatogenic
bacteria themselves and was not provided by external source as reported in other biomimetic studies
**(Rautaray et al., 2003; Han et al., 2018**). In the same manner, **Hou and co-workers, (2011)** found
that the nitrate uptake by *Alternaria* sp. caused of $CaCO_3$ formation via sequestration of respiratory $CO_2$
and thus reduce its emission and indirectly diminishing the rate of global warming.
Finally, the total biological synthesis of calcite and vaterite crystals under nitrate dissimilation
conditions by strains 71A, VIP, and EM4 has not been reported previously. These carbonatogenic
bacteria with their implications in crystal engineering open up the possibility to various prospective
applications include; bioremediation of building stone, monuments/statuary,
consolidation/strengthening of soil/sand, the reduction of the porosity and permeability of geological
formations. Besides, the characteristic features of vaterite [the biocompatibility, biosafety,
biodegradability, high (solubility, porosity, specific surface area), dispersion, accessibility and pH-
sensitivity properties of $CaCO_3$] make it highly appealing in biomedicine applications such as





bone/teeth implants, sensor applications and drug delivery **(Rautaray et al., 2004; Poelvoorde, 2017)**.
Additionally, the overall carbonatogenic process would be utilized in softening of hard water and $CO_2$
capturing from atmosphere and wastewater treatment systems in prospective studies.
**4.  Conclusion**
In conclusion, for the first time the present study demonstrated that the bacterial strains *L. sphaericus*
(71A), *R. planticola* (VIP), and *S. pluricolorescens* (EM4) isolated from Egyptian non-calcareous
niches, induced carbonatogenesis process through nitrate reduction under aerobic and anaerobic
conditions. XRD, EDX and SEM techniques were used to characterize precipitated $CaCO_3$, which
found to differ in their properties according to the type of strain as well as growth conditions.
Precipitated $CaCO_3$ was either calcite or vaterite. Overall, carbonatogenesis process via nitrate
reduction is totally biological, ecofriendly, inexpensive, and promotes $CaCO_3$ precipitations without
accumulation of toxic by-product such as ammonia.
**Data availability.** The data of this study are available for public after a request to the corresponding
author
**Author contributions.** The authors ME, SZ and DA contributed to plan of the wok, results explanation
manuscript writing and data analysis. AK helped in the practical part.
**Competing interests.** The authors declare that they have no conflict of interest.
**Acknowledgment**
This research was conducted at Genetic Engineering and Biotechnology Research Institute, City of
Scientific Research and Technological Applications, Burgelarab city, Alexandria, Egypt
**Financial support.**  This research did not receive any specific grant from funding agencies in the
public, commercial, or not-for-profit sectors.



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





**Figures legends**
**Figure 1:** Neighbour-joining phylogenetic tree based on 16S rDNA gene sequences, illustrating the
relationships between carbonatogenic strains 71A, VIP and EM4 and related species retrieved from
NCBI GenBank, their accession numbers are shown in parentheses. The bootstrap values above 50%,
expressed as percentages of 1000 replications are indicated at the branch points.
**Figure 2:** The morphological differences of calcium carbonate crystals precipitated by carbonatogenic
strains in liquid CCP media; A)-aerobic culture of strain 71A, B)-anaerobic culture of strain 71A, C)-
aerobic culture of strain VIP, D)- anaerobic culture of strain VIP, E)- aerobic culture of strain EM4; F,
G, H, I and K the air dried crystals with the same previous order.
**Figure 3:** Dynamic analysis of carbonatogenesis process associated with changes of pH, $NO_3^-$
concentration, $NO_2^-$ concentration, cell growth, NR activity, EC and $CaCO_3$ weight, mediated by A)-
aerobic culture of strain 71A, B)- anaerobic culture of strain 71A, C)- aerobic culture of strain VIP, D)-
anaerobic culture of strain VIP and E)-aerobic culture of strain EM4. The average of three replica were
performed for each one. To adjust the scale, some parameters are multiplied and/or divided as indicated
on the figures.
**Figure 4:** XRD profile of $CaCO_3$ precipitated by A)- aerobic culture of 71A, B)-anaerobic culture of
71A, C)- aerobic culture of VIP, D)- anaerobic culture of VIP and E)- EM4 culture.
**Figure 5:** EDX crystallographic pattern of $CaCO_3$ precipitated by A)- aerobic culture of 71A, B)-
anaerobic culture of 71A, C)- aerobic culture of VIP, D)- anaerobic culture of VIP and E)- EM4 culture.
**Figure 6:** SEM micrographs of $CaCO_3$ crystal formed by nitrate reducing strains. A, B, C)- crystals
from aerobic culture of 71A, D, E, F)- crystals from anaerobic culture of 71A, G, H, I)- crystals from
aerobic culture of VIP, J, K, L)- crystals from anaerobic culture of VIP and M, N, O)- crystals from
EM4 culture.













**Figure 1**














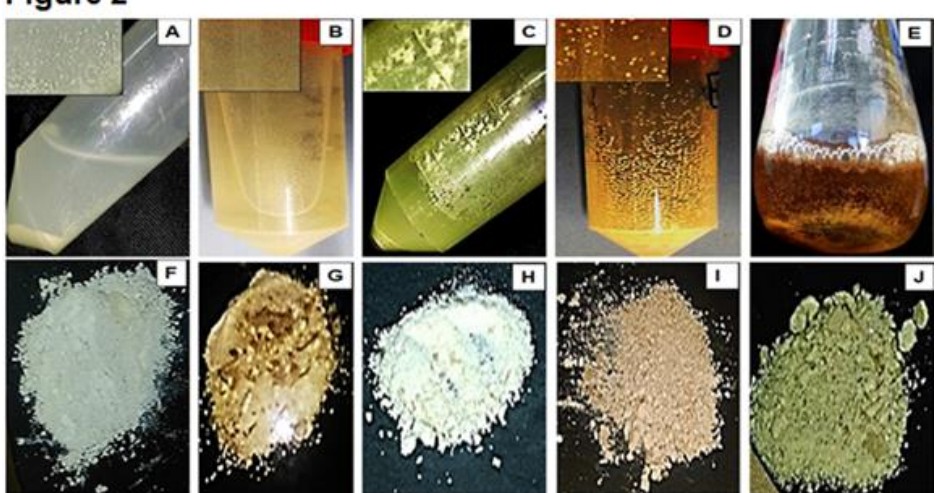

Figure 2
















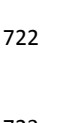

**Figure 3**




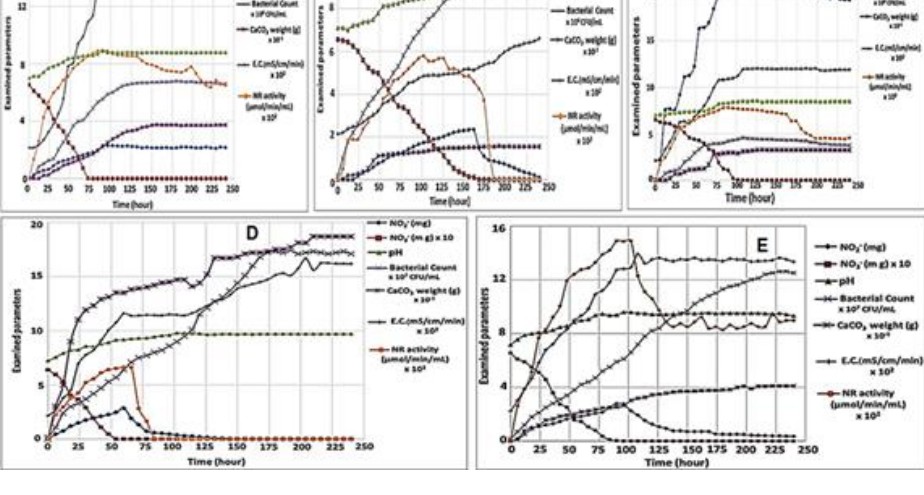


















**Figure 4**

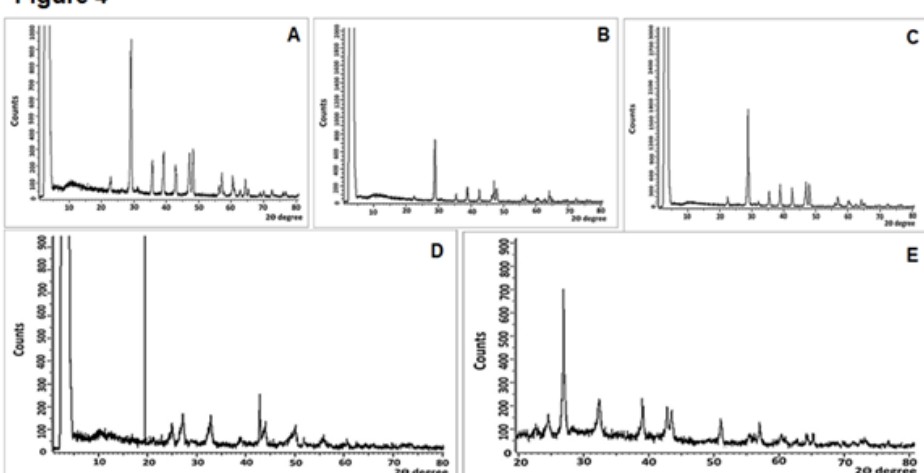



















**Figure 5**

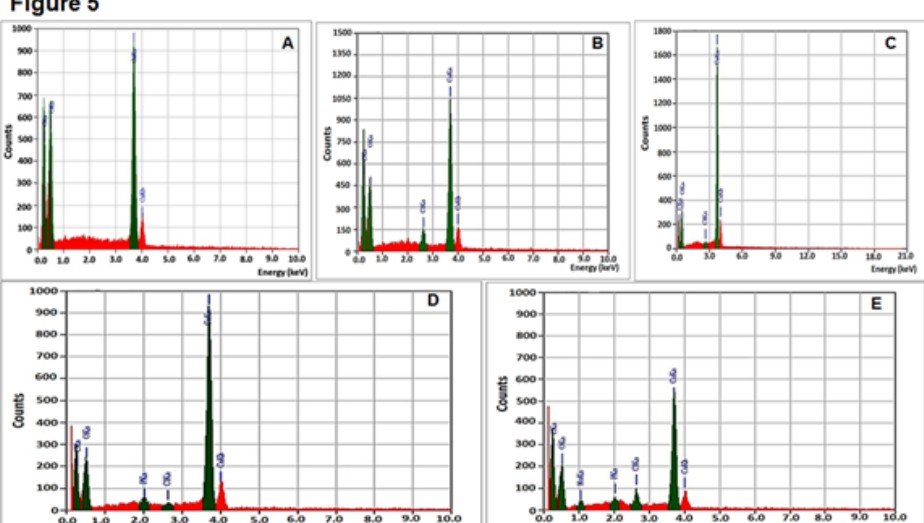



774

775

## Figure 6

