# Peer review of "Calcite and vaterite biosynthesis by nitrate dissimilating bacteria in carbonatogenesis process under aerobic and anaerobic conditions"

_Biogeosciences, 2019_

## Referee Comment (RC1) · Anonymous Referee #1 · 3 Feb 2020

General comments The article deals with an interesting topic that deserves more attention from the scientific community. The manuscript has a potential to be acceptable due to the increasing interest on bioformation of carbonates, both from basic science and technological application point of view. However, in the section "Results and discussion" the authors often refer to articles that are not strictly related to the interpretation of the results obtained, leading to a confused and sometimes inaccurate dissertation. Often, the bibliographic references quoted are nothing but references reported by others authors. Despite the interesting topic dealt with and some results worthwhile to be circulated among the scientific community, the article is extremely confused, badly written, the results and the discussion are often disorganised and difficult to follow.

[Figure]

The discussion is not always coherent with the results reported. I strongly suggest a complete revision by a native English speaker. The manuscript is still far from be ready.

Specific comments In order to help authors to improve their text, I suggest a complete rewrite of the article according to the comments below: 1) Abstract - From line 25 onwards, replace the strain codes (71A, VIP, EM4) with the names of the bacterial species (Lysinibacillus sphaericus, Raoultella planticola, Streptomyces pluricol-ororescens). 2) Keywords - Choose keywords not listed in the title and more relevant to the topic: Lysinibacillus sphaericus, Streptomyces pluricolorescens, Raoultella planticola, CaCO3 bioformation, ...; delete "biocementation".. 3) Lines 45-49 - The different biomineralization processes described in lines 45-49 and 69-77 are reported in a confusing way. Please be clear about BCM, BIM, autotrophic, eterotrophic, SRB, etc. mechanisms that are randomly referred to in the text. 4) Lines 47-49 - References are not strictly related to the statement. 5) Line 51 - I suggest adding the adjective "microbial" to the term "carbonatogenesis". 6) Lines 54-55 - It is not clear how MICCP can participate in the solution of the water crisis. I suggest deleting this sentence. 7) Lines 82-84 - Ureolytic bacteria does not cause the "calcite disintegration", but the "decay of the calcite formation". Thirumalai states: "The use of aerobic bacteria in urea hydrolysis unable to grow in situ due to lack of oxygen, which will results in decay of the calcite formation in time (Van Passen et al., 2010)". 8) Lines 91-92 - Report the increase in carbonate precipitation. 9) Lines 97-98 - In the manuscript there is no experimental evidence about the suitability of CaCO3 crystals for the potential applications listed in the section "Results and discussion" 10) Line 130 - How was the inoculum standardized? 11) Lines 130-137 - How many flasks have been inoculated to carry out what is reported on lines 156-157? Describe the inoculum set more clearly. The flasks analyzed in section "2.5. Study of the parameters associated with CaCO3 precipitation" are the same described in section" 2.3. CaCO3 precipitation and crystals collection"? 12) Line 158: How long were the plates incubated? 13) Line 160 - Report drying times and temperatures of the crystals before being weighed. 14) Line 166 - Have the selected strains been isolated from the same soil? 15) From line

172 onwards - Replace the strain codes with their species names. 16) Lines 172-174: Firmicutes, Bacillaceae, Proteobacteria, Enterobacteriaceae, Actinobacteria, Actinomycetaceae are written in italics. 17) Lines 176-179 - The discussion is not strictly related to the results obtained. 18) Line 131 – please make clear the full composition of M9 media. Without such an information, it is impossible to verify the accuracy in the evaluation of precipitated CaCO3 (lines 238-239) 19) Lines 189-190 – please consider the result obtained by Maciejewska et al. (2017) "Assessment of the Potential Role of Streptomyces in Cave Moonmilk Formation". Front. Microbiol. 8:1181. doi: 10.3389/fmicb.2017.01181 20) From line 208 onward: results are presented and discussed in a very confusing manner preventing the comprehension of the text. 21) The figures are so small and blurry that it is impossible to read them 22) 237-244: the units of measure of the precipitated CaCO3 need to conformed to an unique standard. 23) 247-252: the assumptions made by the authors seem to be of speculative nature. Are they any bibliographic references confirming their interpretation of the results? 24) 283-286: could the detected P derive from the ingredients used to make the culture broth? 25) Please carefully check all the bibliographic references, that are not alway compliant with statement made by the authors. 26) Wei et al. (2015), Wu et al (2011) and APHA, (1999) are not reported in the References section. 27) Lines 241, 436 and 300 and reference section. Please replace "Kaur D.N." con "Dhami N.K." 28) All what was referred to Figures 3, 4, 5 and 6 are not verifiable. The Figures are small, blurry and Illegible. 29) 297-298 and 303: how the authors can state that on Figure 6C and 6E a mucous matrix and mucous substance are evident? 30) 305-309: spores are notoriously inactive and cannot participate to Ca precipitation. 31) Lines 304-309. The statements do not make any sense. The spores in harsh condition remain spores and do nit evolve in vegetative forma, the only one able to contribute to the Ca precipitation. 32) 313-316: the reference Hou et al. (2011) is not coherent with the discussion and it has been quoted in wrong way. 33) 321: Rodriguez-Navarro et Al. (2012) obtained calcite and vaterite precipitation under experimental conditions widely differing from those described in this manuscript 34) 327-328:please drop out.

The sentence is useless in this context 35) 335-483: the discussion is vague and superficial and carried out in a confusing way. In addition, the references quoted refer to article that used experimental protocols widely differing from those described in this manuscript 36) Figure 6: The pictures are too small and difficult to read. Please consider a reduction of the photos by eliminating the redundant ones.

Please also note the supplement to this comment:
https://www.biogeosciences-discuss.net/bg-2019-444/bg-2019-444-RC1-supplement.pdf

---

## Referee Comment (RC2) · Anonymous Referee #2 · 3 Feb 2020

The topic of the article is in itself aligned with the scope of BG. Unfortunately, the image quality (resolution) of the figures is so poor that axes and legends are illegible, in particular for Figures 3-5. Therefore, it is not possible to judge whether the experimental data are of sufficient quality. At a first glance, the text of this manuscript is poorly written and confusing (see anonymous reviewer #1 for more details). I strongly suggest the authors to consider the more detailed suggestions of reviewer #1 during a major overhaul of the manuscript contents. Furthermore, higher resolution images are needed for all figures and a thorough revision by a native English speaker is necessary before consideration and discussion of this manuscript is feasible.

---

## Referee Comment (RC3) · Anonymous Referee #3 · 26 Feb 2020

General comments The article (bg-2019-444) explores the biogenically induced carbonate precipitation triggered by nitrate reducing bacteria in both oxic and anoxic environments. Three types of nitrate reducing bacteria were isolated from three different, non-calcareous sampling sites in Egypt, specified, and incubated in 200 ml anoxic and oxic incubation media at 30°C for 10 days. Two incubation media without bacterial cultures served as references. During that time different parameters like pH, nitrate and nitrite concentrations, bacterial count, electrical conductivity, and nitrate reductase activity were analyzed. The deposited CaCO3 amount was quantified, the type and morphology were investigated by XRD, EDX, and SEM analysis. The present study demonstrates that (I) the biogenically induced carbonate precipitation by nitrate reduc-

ing bacteria is more pronounced under during aerobic growth than during anaerobic conditions and that (II) the type (in this case calcite and vaterite) and morphology of the formed carbonate is controlled by species and growth conditions. The study represents a significant contribution to the research on carbonate formation and provides a potential application in the carbonate-processing industry. It addresses a relevant research field within the scope of BG, however, there are major aspects that need to be worked on before considering a publication: Specific comments The manuscript could benefit from a language check. Some sentences are almost impossible to understand. The partly confusing text structure can significantly be improved. There are many literature references distributed in the results and discussion chapter that have little relation to the text before and after (for instance, Lines 174-179, 203-207, 298-300, 313-316). I don't see how these sentences support the discussion or how they can lead to the conclusions. Further, I couldn't read the figure because of the poor quality. Figure 3: I would suggest to add two diagrams showing the parameters of the anoxic and oxic control media, present all diagrams on the same size, and use the same colors for the same parameters. Line 95: If those bacteria support CaCO3 precipitation, why was it possible to isolate them from a non-calcareous habitat? Chapter 2.3: this chapter describes the experimental incubation conditions; I think this should be reflected in the title. Line 134: maybe a short description about the anoxic incubation set-up? Chapter 2.5 this chapter should be merged with chapter 2.3 Line 174-179: what does this information have to do with identification and isolation of the studied bacteria? These sentences seem lost there. Line 191: this title is too general. The whole paper is about CaCO3 deposition. This chapter rather describes the relationship between NR, oxic conditions and biogenically induced carbonate precipitation. Line 211-213: this is an important general observation and explanation that is followed by description of the results. This text structure should be changed. Line 226-228: I don't understand this sentence. Also, a new paragraph starts here. The description and the discussion of the results is mixed up throughout this chapter. I would suggest to change that. Line 240-241: How is this information about a completely different microbe related to the

former sentence? How does this information help you to get to your conclusions? I think this sentence, as it stands there, is not necessary. Line 243-245: How much sense does it make to compare the CaCO3 precipitation amounts (in different units!) from three completely different experiments? What can we learn from that? It is not productive to only list results from different experiments if no conclusion can be drawn from that. Chapter 3.4: I would recommend to further subdivide this chapter because this helps to build a proper text structure. Line 298-300: How does this notion of another study stand in relationship to the sentences before and after? How does it impact your discussion? I don't think this information is helpful here. Line 313-316: Another notion of a further study. How much sense does this short notion about a study with a fungus species make? How does this information help your argumentation? Line 320: what exactly was the finding of Li et al. (2012)? Line 321-323, 326-328: see comments above (Line 298-300, 313-316) Line 331: why the reference? Isn't this the conclusion of your study? Line 375-376: see comments above (Line 298-300, 313-316) Line 377-379: see comments above (Line 298-300, 313-316) Line 384-385: this sounds like a great conclusion. Line 490: maybe sum up likely reasons for the formation of the different CaCO3 types. Technical corrections Line 50: "microbial", not "Microbial" Line 54: confusing reference Line 67: do you mean microorganisms with different metabolisms? Line 70: What exactly seems to be more abundant? Line 98: CaCO3 crystal Line 106: where all three bacteria found in all three sampling sites? Line 118: 1 $\mu$mol Line 121: using 16S. . . Line 122: what is PCR? Line 123: "in" instead of "elsewhere" Line 124: what does BLAST mean? Line 131: What is a M9 media? Line 136: 10,000 g Line 143: delete one parenthesis Line 144: operates Line 163: What is APHA? Line 175: What is MICCP? Line 189: please rewrite. Line 192: please add the reference after "preceding literature" Line 195: red semicolon Line 197: different appearances: the strains or the crystals? Please be more specific. Line 198: please add parentheses around (Fig. 2) Line 201: delete "were". Appeared in all anaerobic cultures? Also, the comparison between irregular crystals in the anaerobic cultures and fine white powder in the supposedly aerobic culture appears strange. The former describes the morphology of crystals, the latter can roughly tell something about the crystal size. Line 203: "...brown aggregated pellets." Do you speak about the aerobic culture? Line 209: were instead of "...was monitored." Line 249: "It is worth to mention..." Why is it worth to mention? Line 253: "by almost all of the parameters"? Line 255-258: this is difficult to understand Line 277: are Line 281: "...EDX peaks that..." Line 286-288: reference is missing Line 292: was instead of were Line 352-354: please rewrite. Line 358-359: reference is missing. Line 359-360: either delete or elaborate further. Line 363: "...is controlled by several factors." Line 367: this sentence is incomplete. Line 387: cations are always positively charged. Line 451-453: this sentence is incomplete. Line 453: better: vaterite formation. Line 465: better for what? Line 466-467: confusing sentence.

―――――――――――――――――

---

## Author Comment (AC1) · 18 Mar 2020

Response to peer-reviewers Firstly, we appreciate the reviewers' feedbacks and their careful reading of our manuscript, interest in our study and thoughtful comments that greatly improve the quality of the paper. Secondly, we did our best to respond to the points raised. The Referees have brought up some constructive suggestions and we appreciate the opportunity to clarify our research objectives and results. As indicated below, we have checked all the general and specific comments pointed out and have made the necessary changes accordingly to their indications.

• Reviewer #2: The topic of the article is in itself aligned with the scope of BG.

[Figure]

C1)-Unfortunately, the image quality (resolution) of the figures is so poor that axes and legends are illegible, in particular for Figures 3-5. Therefore, it is not possible to judge whether the experimental data are of sufficient quality. At a first glance, the text of this manuscript is poorly written and confusing (see anonymous reviewer #1 for more details). C2)-I strongly suggest the authors to consider the more detailed suggestions of reviewer #1 during a major overhaul of the manuscript contents. Furthermore, C3)-higher resolution images are needed for all figures and a thorough revision by a native English speaker is necessary before consideration and discussion of this manuscript is feasible.

A1- Thanks for your deep notice. We fixed and adjusted all figures to be readable and informative.

A2- As mentioned previously, we followed your respective opinion and also the opinion of respective Reviewer # 1 and # 3. We revised the manuscript thoroughly and additionally, English native speaker checked English quality of revised manuscript.

A3- We followed your suggestion in revised form of manuscript

―――――――――――――――――――

---

## Author Comment (AC2) · 18 Mar 2020

Response to peer-reviewers Firstly, we appreciate the reviewers' feedbacks and their careful reading of our manuscript, interest in our study and thoughtful comments that greatly improve the quality of the paper. Secondly, we did our best to respond to the points raised. The Referees have brought up some constructive suggestions and we appreciate the opportunity to clarify our research objectives and results. As indicated below, we have checked all the general and specific comments pointed out and have made the necessary changes accordingly to their indications. • Reviewer #3:

Specific comments:

[Figure]

C1)-The manuscript could benefit from a language check. Some sentences are almost impossible to understand. The partly confusing text structure can significantly be improved.

A1: We followed your respective suggestion and we revised the manuscript thoroughly and additionally, English native speaker checked English quality of revised manuscript. We rewrote the manuscript and totally reorganize its structure, as well, in particular the results and discussion section.

C2)-There are many literature references distributed in the results and discussion chapter that have little relation to the text before and after (for instance, Lines 174-179, 203-207, 298-300, 313-316). I don't see how these sentences support the discussion or how they can lead to the conclusions.

A2: All references were revised regarding to their suitability and fitness to the results or their interpretations.

 c For lines 174-179: We referred to different bacterial groups (phylum Firmicutes, family Bacillaceae; phylum Proteobacteria, family Enterobacteriaceae and phylum Actinobacteria, family Actinomycetaceae) that related to the exact classification of our isolates and had the same scope ($CaCO_3$ production with various applications). So, we thought the importance for mentioning other studies that are similar to our results and support our vision. However, according to your notice and Reviewer's #1 recommendation, we stated such information in convenient way in the revised manuscript (line 227-230, page 10), particularly that there were different publications stated this point and dealt with their data with the same sort of discussion.

 c For lines 203-207: We would manifest that we obtained $CaCO_3$ crystal with different colors and morphology by different bacterial species, so such variations were observed due inter-species differences. The exact observation was recorded by Montano-Salazar et al., (2017), who found that, Rhodococcus qingshengii M101 produce spherical brown crystals; Arthrobacter crystalopoyetes produced irregular yellowish crystals

and Psychrobacillus psycrodurans showed irregular white/beige crystals of CaCO3. Such finding supports our results which were listed at page 12 (lines 258-262).

• For lines 298-300: Caicedo-Pineda et al., (2018) mentioned that the highly pure CaCO3 give EDX spectrum with Ca peak higher than C peak. That comes in agreement with our result. As observed in Figure 5, the peaks intensities of Ca are more than peak intensities of C in all examined samples, which finally implied pure CaCO3.

• For lines 313-316: From our point of view, this reference stated that the size of the precipitated calcite by Alternaria sp. was ranged from less than 1- and not exceeded 10 $\mu$m, which was agreed with the size of calcite crystals formed by strain Raoultella planticola (VIP), and under the same nitrate utilization conditions. Despite that, we followed your notice and the opinion of Reviewer #1 and deleted it in the revised manuscript.

C3)- Further, I couldn't read the figure because of the poor quality.

A3: We apologize for such unintended mistake. Respect your opinion. All figures were adjusted and subjected to improve the image quality using photo editing software. All figures become clear enough to read and perceive.

C4)-Figure 3: I would suggest to add two diagrams showing the parameters of the anoxic and oxic control media

A4: Respect your opinion. However, adding such diagrams with all parameters which will remain constant along 10 days incubation would not reveal any additional observation. Therefore, presence of such diagrams with fixed parameters will not add information for comparing or interpretation of data. Rather, it would minimize the figures sizes of the other experiments which contained important data about the process.

C5)-present all diagrams on the same size, and use the same colors for the same parameters.

A5: All diagrams at figure 3 were in the same size. All the same parameters had the same and unique colors/mark. For example: NO2- is blue line with rhombus mark,

NO3- is red line with square mark, pH green line with triangular mark, etc. The lateral caption illustrated such details.

C6)-Line 95: If those bacteria support CaCO3 precipitation, why was it possible to isolate them from a non-calcareous habitat?

A6: We focused our study on nitrate reduction mechanism and its main role in CaCO3 precipitation under oxic/anoxic conditions. So, during screening and isolation process, we concentrate on finding organisms with such mechanism to ensure their validity in the CaCO3 precipitation. On the other hand, the inverted pathway, screening from calcareous area ensures isolation of CaCO3 precipitating organisms, but didn't ensure presence of nitrate reduction mechanism; they might exhibit CaCO3 precipitation through urea degradation, deamination of proteins, carbonyle anhydrase or sulfate reduction (listed at introduction section).

C7)-Chapter 2.3: this chapter describes the experimental incubation conditions; I think this should be reflected in the title.

A7: Thanks for your suggestion. We followed it and added this note in the title (Page 7 & Page 11, Aerobic/ anaerobic CaCO3 precipitation).

C8)-Line 134: maybe a short description about the anoxic incubation set-up?

A8: Thanks for your suggestion. We believed that it is well known and reported tremendously in other literatures. Despite, we mentioned what you suggested (Line 165-168; Page 8).

C9)-Chapter 2.5 this chapter should be merged with chapter 2.3.

A9: according to your recommendation and also Reviewer's #1 note, we merged both sections and clarified the studied items (Line 157, page 7).

C10-Line 174-179: what does this information have to do with identification and isolation of the studied bacteria? These sentences seem lost there.

A10: A17: We referred to different bacterial groups (phylum Firmicutes, family Bacillaceae; phylum Proteobacteria, family Enterobacteriaceae and phylum Actinobacteria, family Actinomycetaceae) that related to the exact classification of our isolates and had the same scope (CaCO3 production with various applications). So, we thought the importance for mentioning other studies that are similar to our results. However, according to your notice, we stated such information in brief and in a convenient way in the revised manuscript (line 227-230, page 10), particularly there were different publications stated this point and dealt with their data with the same sort of discussion.

C11- Line 191: this title is too general. The whole paper is about CaCO3 deposition. This chapter rather describes the relationship between NR, oxic conditions and biogenically induced carbonate precipitation. A11: We are grateful for your note. In the revised manuscript, this section was divided into two parts with two different titles (3.3- Aerobic / anaerobic CaCO3 precipitation) which confirmed deposition of CaCO3 crystals and described appearance differences between precipitated crystals under aerobic/anaerobic conditions. The other title is (3.4- Study of the parameters associated with CaCO3 precipitation), which described the parametric changes during CaCO3 deposition process under aerobic/anaerobic nitrate reduction conditions. C12-Line 211-213: this is an important general observation and explanation that is followed by description of the results. This text structure should be changed. A12: We are grateful for your recommendation. The changes were performed (Line 329-334, page 15, revised manuscript). C13-Line 226-228: I don't understand this sentence. Also, a new paragraph starts here. A13: We followed your recommendation and start new paragraph at such point. As observed in Figure 3, under aerobic conditions, periplasmic nitrate reductase was detected along 10 days of incubation even after complete depletion of NO3- (nitrate reductase substrate). While, anaerobically, membrane-bound nitrate reductase was detected only in the presence of NO3- (for 180 and 70 h for L. sphaericus and R. planticola, respectively). That means, both strains have two different types of nitrate reductases which were expressed under different aeration conditions and both have different physiological roles as indicated in the manuscript. The structure of sen-

tence was improved to be more understandable and clearer (Line 284-296, page 13).

C14-The description and the discussion of the results is mixed up throughout this chapter. I would suggest to change that. A14: We are grateful for your recommendation. Reorganization and rearrangement of this section were performed. C15-Line 240-241: How is this information about a completely different microbe related to the former sentence? How does this information help you to get to your conclusions? I think this sentence, as it stands there, is not necessary.

A15: Firstly, we would like to declare that there was no previous report about our examined organisms in CaCO3 deposition using nitrate reduction strategy, till our knowledge, so there was no more data available on the same organisms could be compared with our results. Secondly, we compare the overall process with its output (CaCO3 formation) and such comparison includes time frame of formation, type of organism, polymorph, size of crystals and amount of precipitated CaCO3. We referred to such data at their proper position in manuscript. Where, at SEM we referred to the shape and size; at XRD we referred to the polymorphs. Whereas, at parametric changes study, we referred to time frame and amount of deposited crystals. Thus, our results are characteristic and strains produced considerable amount of CaCO3 upon comparison with cited strains.

C16-Line 243-245: How much sense does it make to compare the CaCO3 precipitation amounts (in different units!) from three completely different experiments? What can we learn from that? It is not productive to only list results from different experiments if no conclusion can be drawn from that.

A16: According to your valuable note, we made the required changes to confirm uniformity of unites according to standards (expressed the data in gm/ 100 mL). Herein, we thought that it is logic to compare the whole process (CaCO3 precipitation) and its output with other works. It is true that are completely different strategies (ureolysis & nitrate reduction), but the same target was obtained. So, we displayed the performance

of our strains under such conditions, compared the metabolic activity with expressed enzyme, and eventually compared the product either amount or polymorph, which all such data fall in the circle of CaCO3 precipitation. Our main conclusion is the possibility of production of different polymorph of CaCO3 with considerable amount and also within suitable frame of time, which will provide good impact at technological, environmental and medical levels. Such conclusion comes from comparison with other studies output.

C17-Chapter 3.4: I would recommend to further subdivide this chapter because this helps to build a proper text structure. A17: We are grateful for your recommendation. We followed it and subdivided into 3 sections (XRD (page 16), EDX (Page 19) and SEM (Page 20). C18- Line 298-300: How does this notion of another study stand in relationship to the sentences before and after? How does it impact your discussion? I don't think this information is helpful here. A18: We pointed out to other studies which either agreed or contrast our finding. This sentence supports our finding, in which the same shape of crystals with presence of obvious extracellular polymeric substances (EPS) was deposited. The presence of EPS is a characteristic feature. As mentioned above, the whole process with its characteristic output was compared regardless the producing organism as long as in the exact target of study. So, we thought that sentence present in a coherent way with the previous one. C19-Line 313-316: Another notion of a further study. How much sense does this short notion about a study with a fungus species make? How does this information help your argumentation? A19: From our point of view, this reference stated that the size of the precipitated calcite by Alternaria sp. was ranged from less than 1- and not exceeded 10 $\mu$m, which was agreed with the size of calcite crystals formed by strain Raoultella planticola (VIP), and under the same nitrate utilization conditions. Overall, we quoted this study as fall in the same scope (CaCO3 deposition) regardless the producing organism. Finally, we take your point of view in our consideration and removed it. C20-Line 320: what exactly was the finding of Li et al. (2012)? A20: Such reference highlighted to the presence of bacterial imprint on the surface of CaCO3 crystals in the formed of holes, which is agreed with our results.

Li and coworkers attributed such holes to the bacterial cells which act as nucleation site. The sentence was rewritten to be more clear (Line 479, page 21). C21- Line 321-323, 326-328: see comments above (Line 298-300, 313-316) A21: All sentences were revised and our reply indicated above. Line 321-323 (the answer of Reviewer's #1, C33, page 11). Line 326-328 (the answer of Reviewer's #1, C34, page 12). C22-Line 331: why the reference? Isn't this the conclusion of your study? A22: It is the conclusion of the study. The reference was deleted. The whole structure of this section (Results & discussion) was reformatted. C23-Line 375-376: see comments above (Line 298-300, 313-316) A23: In this sentence, we displayed reference that mentioned different ranges of pH at which calcite and vaterite were formed. Subsequently, this sentence boosts our finding. C24-Line 377- 379: see comments above (Line 298-300, 313-316) A24: In this sentence, we displayed the opposite finding to our study. We intended to display the diversity of conditions to obtain different polymorphs. Nonetheless, we deleted such sentence. C25-Line 490: maybe sum up likely reasons for the formation of the different $CaCO_3$ types. A25: Agreed with you. C26-Technical corrections Line 50: "microbial", not "Microbial". A26: Thanks for your note. The correction was performed (line 63, page 3). C27-Line 54: confusing reference A27: Thanks for your note. The sentence with reference was revised and the correction was performed (line 68, page 4). C28-Line 67: do you mean microorganisms with different metabolisms? A28: We meant different microbial group could synthesize different $CaCO_3$ polymorph (calcite, vaterite, aragonite, etc.). The sentence became clear after amendment and revision (Line 84, page 4) C29-Line 70: What exactly seems to be more abundant? A29: The heterotrophic one is more abundant. To prevent confusion and misunderstanding, we rewrote the sentence (line 87, page 4). C30-Line 98: $CaCO_3$ crystal A30: Thanks for your note. The correction was performed (line 123, page 6). C31-Line 106: where all three bacteria found in all three sampling sites? A31: Thanks for your note. We referred to such point in section 3.1 (line 214-216, page 10). C32-Line 118: 1 $\mu$mol A32: The correction was performed (line 145, page 7). C33-Line 121: using 16S. A33: The correction was performed (line 149, page 7). C34-Line 122: what is PCR?

A34: Polymerase chain reaction. It is well-known and the most widely used molecular technique. It is applied in identification and differentiation purposes. C35-Line 123: "in" instead of "elsewhere" A35: The correction was performed (line 152, page 7). C36-Line 124: what does BLAST mean? A36: In bioinformatics, BLAST (basic local alignment search tool) is an algorithm and program for comparing primary biological sequence information, such as the amino-acid sequences of proteins or the nucleotides of DNA and/or RNA sequences. By such tool, query (protein or nucleotide sequence) was compares with library or database of sequences, and identifies library sequences that resemble the query sequence above a certain threshold. C37-Line 131: What is a M9 media? A37: The media for precipitation. According to your opinion and Reviewer's #1 opinion, the composition of media was listed (Line 162-164; Page 7-8). C38-Line 136: 10,000 g A38: The correction was performed (line 190, page 9). C39-Line 143: delete one parenthesis A39: The correction was performed (line 203, page 9). C40-Line 144: operates A40: The correction was performed (line 204, page 9). C41-Line 163: What is APHA? A41: American Public Health Association (APHA, 1999). It provides Standard Methods for the Examination of Water and Wastewater. C42-Line 175: What is MICCP? A42: Microbial induced calcium carbonate precipitation (paragraph 1, page 3) C43-Line 189: please rewrite. A43: The correction was performed (line 240-241, page 11). C44-Line 192: please add the reference after "preceding literature" A44: The reference was added (line 245-246, page 11). C45-Line 195: red semicolon A45: The correction was performed (line 247, page 11). C46-Line 197: different appearances: the strains or the crystals? Please be more specific. A46: We meant crystals. The correction was performed (line 249, page 11). C47-Line 198: please add parentheses around (Fig. 2) A43: The add parentheses around (Fig. 2) was added (line 250, page 11). C48-Line 201: delete "were". A48: The correction was performed (line 255, page 12). C49-Appeared in all anaerobic cultures? A49: Yes, appeared in all anaerobic cultures. We declared this point in the revised manuscript (Line 256, page 12). C50-Also, the comparison between irregular crystals in the anaerobic cultures and fine white powder in the supposedly aerobic culture appears strange. The former describes

the morphology of crystals, the latter can roughly tell something about the crystal size. A50: Both expressions described general appearance (color, texture and size). We reported equal description for both types. Where, large and coarse beige or buff color and irregular crystals (anaerobically) were compared to fine white powder (aerobically). The description involved size (large coarse and fine), color (beige/buff and white) and texture (powder and crystal). Generally, any salt would be described morphologically either powder or crystal in chemistry point of view. The exact morphology and exact size were determined through SEM, which confirmed such optical observation. C51-Line 203: ". . .brown aggregated pellets." Do you speak about the aerobic culture? A51: Yes, such point (aerobic culture of EM4) was manifested previously in section 3.2 (Line 239-241, page 11). C52-Line 209: were instead of ". . .was monitored." A52: The correction was performed (line 265, page 12). C53-Line 249: "It is worth to mention. . ." Why is it worth to mention? A53: We thought that it is important to mention such information, where, EC used mainly in determination of the ureolytic efficiency in MICP and didn't use before in determination of nitrate reduction performance. So, we stated that it is general method could be used to give an insinuation about metabolic activity and overall mineralization process and it is worked and gave the required information. C54-Line 253: "by almost all of the parameters"? A54: Thanks for your note, we amended the sentence. Where, most of examined parameters and not all exhibited such observation. Because, there are some parameters were consumed completely before the end of experiment (10 days) such as NO3- (in aerobic and anaerobic conditions), NR & NO2- (anaerobic conditions). So, the rest of parameters exhibited such slight decline or stability state (line 354, page 16). C55-Line 255-258: this is difficult to understand A55: The sentence was revised to be easier for understanding (line 355-360, page 16). C56-Line 277: are A56: The correction was performed (line 423, page 19). C57-Line 281: ". . .EDX peaks that. . ." A57: The correction was performed (line 430, page 19). C58-Line 286-288: reference is missing A58: The reference was added (line 437, page 19). C59-Line 292: was instead of were A59: The correction was performed (line 448, page 20). C60-Line 352-354: please rewrite. A60: The structure of

this point was completely reformatted. Some sentences were deleted. C61-Line 358-359: reference is missing. A61: This sentence is the result of our study, so no reference was added. Additionally, the structure of this point was completely reformatted. C62-Line 359-360: either delete or elaborate further. A62: This sentence is quote that supports our results. The quoted reference didn't mention the exact size of crystals at both cases regarding strain of L. sphaericus, to elaborate further. For clarification, the shaking conditions provides more aeration and homogeneity than static one (it is known information), so we quoted it as a reference for comparison between different levels of aeration (Line 483-487, page 21). C63-Line 363: ". . .is controlled by several factors." A63: The sentence was amended according to your recommendation (lines 380, page 17). C64-Line 367: this sentence is incomplete. A64: The sentence was revised (line 385-387, page 17). C65-Line 387: cations are always positively charged. A65: The whole manuscript was revised and rewrote and some sentences were adjusted or deleted. C66-Line 451-453: this sentence is incomplete. A66: The sentence was revised (line 497-499, page 22). C67- Line 453: better: vaterite formation. A67: The sentence was amended according to your recommendation (line 496, page 22). C68-Line 465: better for what? A68: For synthesis process and subsequent application. The natural synthesis methods and naturally produced products are always safe, biocompatible and ecofriendly. Any natural product is better than any synthetic one. Where, natural products were free from any traces of reactant substances. Such trace could cause undesired impact.

C69-Line 466-467: confusing sentence. A69: The sentence meant that the produced CO2 and subsequently CO3- are produced from the metabolic activity of bacteria and from their respiration, so it is truly biogenic process. Biomimetic process includes employment and addition of biologically origin molecules such as proteins and polymers in a synthetic reaction.

---

## Author Comment (AC3) · 18 Mar 2020

Response to peer-reviewers Firstly, we appreciate the reviewers' feedbacks and their careful reading of our manuscript, interest in our study and thoughtful comments that greatly improve the quality of the paper. Secondly, we did our best to respond to the points raised. The Referees have brought up some constructive suggestions and we appreciate the opportunity to clarify our research objectives and results. As indicated below, we have checked all the general and specific comments pointed out and have made the necessary changes accordingly to their indications.

The reviewer's comments are in Bold and underlined

• Reviewer #1: General comments

The article deals with an interesting topic that deserves more attention from the scientific community. The manuscript has a potential to be acceptable due to the increasing interest on bioformation of carbonates, both from basic science and technological application point of view. However, (1) in the section "Results and discussion" the authors often refer to articles that are not strictly related to the interpretation of the results obtained, leading to a confused and sometimes inaccurate dissertation. Often, the bibliographic references quoted are nothing but references reported by others authors. Despite the interesting topic dealt with and some results worthwhile to be circulated among the scientific community, (2) the article is extremely confused, badly written, the results and the discussion are often disorganised and difficult to follow. The discussion is not always coherent with the results reported. (3) I strongly suggest a complete revision by a native English speaker. The manuscript is still far from be ready.

A1: Thanks for your notice and deep revision. We have carefully revised the manuscript and the references cited. In the revised manuscript the references were more fit to the discussion and interpretation.

A2: Thanks for your comments. We have carefully revised the manuscript and we have taken special care to clarify our results in narrative way and with convenient interpretations. We would clarify that each point studied in this research was essential to make the research complete work and free from defect or shortage, as much as we can. Any details or discussion in each result was employed to interpret the result concisely, precisely and without redundancy. Thorough revision and total reorganization for the manuscript were performed.

Where, it began by selection of the most potent strains in both NR activity and also in CaCO3 precipitation. The selected isolates were identified and phylogenetic tree for them was constructed, interpreted and compared with other studies briefly (Page 10). Then NR activity of selected strains and their CaCO3 deposition capability were determined aerobically and anaerobically in a comparative way (Page 11). Thereafter, the complete process for CaCO3 deposition under oxic/anoxic nitrate utilization conditions was studied in details and also in comparative way which was not studied before, till our knowledge (Page 11-16). The variation in the size, morphology and the identity of crystals was revealed by mineralogical analysis EDX, SEM and XRD. The reasons for different polymorph (calcite/vaterite) (Page 17-18), different size and morphology (Page 20-21) were elucidated. Finally, the comparison with other literatures was performed in a way that served the manuscript.

A3: We appreciate your suggestion. We do our best in this concern. Correction of grammatical errors and improvement for English quality were carried out by a native English-speaking colleague as suggested and additionally by expert website. The suggested corrections have been made.

Specific comments

In order to help authors to improve their text, I suggest a complete rewrite of the article according to the comments below:

C1) Abstract - From line 25 onwards, replace the strain codes (71A, VIP, EM4) with the names of the bacterial species (Lysinibacillus sphaericus, Raoultella planticola, Streptomyces pluricolororescens).

A1: The correction was performed according to your suggestion.

C2) Keywords - Choose keywords not listed in the title and more relevant to the topic: Lysinibacillus sphaericus, Streptomyces pluricolorescens, Raoultella planticola, CaCO3 bioformation, ...; delete "biocementation"..

A2: Appreciating your recommendation. Your suggestions were considered Page 2).

3) Lines 45-49 - The different biomineralization processes described in lines 45-49 and 69-77 are reported in a confusing way. Please be clear about BCM, BIM, autotrophic, eterotrophic, SRB, etc. mechanisms that are randomly referred to in the text.

A3: we are grateful for your comment. Your suggestions were considered and we manifested the detailed difference between BIM and BCM clearly in the revised manuscript (Line 50-60, page 3). For autotrophic, heterotrophic, SRB, etc., they are the pathways by which microbes deposit CaCO3 in microbial induced calcium carbonate precipitation process (MICCP). Generally, BIM was mediated by those mechanisms (Line 86, page 4).

4) Lines 47-49 - References are not strictly related to the statement.

A4: Thanks for your deep revision. We deleted (Ghosh et al., 2019) only. With deep reviewing, we confirmed that the other references mentioned both types of biomineralization.

5) Line 51 - I suggest adding the adjective "microbial" to the term "carbonatogenesis".

A5: We followed your recommendation (Line 63, page 3).

6) Lines 54-55 - It is not clear how MICCP can participate in the solution of the water crisis. I suggest deleting this sentence.

A6: We followed your recommendation and deleted it.

7) Lines 82-84 - Ureolytic bacteria does not cause the "calcite disintegration", but the "decay of the calcite formation". Thirumalai states: "The use of aerobic bacteria in urea hydrolysis unable to grow in situ due to lack of oxygen, which will results in decay of the calcite formation in time (Van Passen et al., 2010)".

A7- The sentence was corrected (line 103-106, Page 5).

C8) Lines 91-92 - Report the increase in carbonate precipitation.

A8- The value was added (Line 115, page 6).

C9) Lines 97-98 - In the manuscript there is no experimental evidence about the suitability of CaCO3 crystals for the potential applications listed in the section "Results and

discussion"

A9- We would point out that the current study is the basic stone for several applications. Through such study, the characteristic features of bioformed CaCO3 such as their shape, purity (single or mixed polymorphic phase), size and time frame, at which they formed, were recognized. All such features determined which strain and under which condition could be employed in which application.

In addition, the general characteristic features of calcite and vaterite were known previously from other literatures (Tas, 2009; Trushina et al., 2014; Dizaj et al., 2015; Svenskaya et al., 2016). Where, calcite is the most stable form, potent and the least solubility. So, it could be harnessed in applications required stabilization and low dissolution, such as soil consolidation and sequestration of pollutants (e.g. heavy metals). While, vaterite is metastable and rapidly dissolves at acidic pH; thus, it can undergo degradation both in vitro and in vivo solutions like body fluid which contains a number of acidic metabolites, such as citrate, lactate and acid hydrolysis enzymes. So, the spherical shaped vaterite formed biologically by our strains Raoultella planticola and Streptomysetes pluricolorescens could be utilized in medical applications.

As indicated by mineralogical analysis, the biosynthesized calcite by Lysinibacillus sphaericus (fine and large particles) would be harnessed in strengthening of soil/sand (according to particles size), crack healing, and the reduction of the permeability of geological formations. Currently, in our lab, there is an ongoing study for remediation of heavy metals using Raoultella planticola under aerobic & anaerobic nitrate utilization. Additionally, the vaterite bioformed by Streptomysetes pluricolorescens is invested nowadays, in our lab, medically in drug delivery.

• Svenskaya, Y.I., Fattah, H., Inozemtseva, O.A., Ivanova, A.G., Shtykov, S.N., Gorin, D.A., and Parakhonskiy, B. V.: Key Parameters for Size- and Shape-Controlled Synthesis of Vaterite Particles, Crystal Growth Design, 1, 331-337

• Trushina D., Tatiana V. Bukreeva, Mikhail V. Kovalchuk, Maria N. Antipina, CaCO3

vaterite microparticles for biomedical and personal care applications, Materials Science & Engineering C (2014),

• Dizaj S., Mohammad Barzegar-Jalali, Mohammad Hossein Zarrintan, Khosro Adibkia, Farzaneh Lotfipour, 2015. Calcium Carbonate Nanoparticles; Potential in Bone and Tooth Disorders. Pharmaceutical Sciences, March 2015, 20, 175-182.

• Tas A., 2009. Monodisperse Calcium Carbonate Microtablets Forming at 701C in Prerefrigerated CaCl2–Gelatin–Urea Solutions. Int. J. Appl. Ceram. Technol., 6 [1] 53–59.

C10) Line 130 - How was the inoculum standardized?

A10: We use fixed amounts of inoculum according to Mcfarland standard, in association to bacterial count, to ensure uniformity of the inoculum. As reported at Material & methods section (2.3) line 161 (page 7), About 250 $\mu$L of bacterial cultures (1.8 x 106 CFU/mL) was used as inoculum for precipitation test.

C11) Lines 130-137 - How many flasks have been inoculated to carry out what is reported on lines 156-157? Describe the inoculum set more clearly. The flasks analyzed in section "2.5. Study of the parameters associated with CaCO3 precipitation" are the same described in section" 2.3. CaCO3 precipitation and crystals collection"?

A11: We inoculated 10 flasks for each strain under each condition to carry out what reported in point 2.3 (lines 159-186 (page 7- 9), revised manuscript and lines 130-137 in previous manuscript). According to your suggestion, we added the details about inoculum set and experiment design (Line 176-186, page 8-9).

For the points (2.3 & 2.5 in old manuscript), the exact media, inoculation, incubation conditions and incubation time were applied exactly, but the studied items were different. So, according to your note and also Reviewer's #3 suggestion, we merged both sections and clarified the studied items.

12) Line 158: How long were the plates incubated?

A12: The plates were incubated for 24 h (line 188, page 9 (revised manuscript; Line 158 in old manuscript).

13) Line 160 - Report drying times and temperatures of the crystals before being weighed.

A13): We are grateful for your notice. The required data were added (Line 192, page 9).

14) Line 166 - Have the selected strains been isolated from the same soil?

A14: The samples were collected from different non-calcareous Egyptian sites (different governates), as reported (line 131-133, Page 6). Additionally, according to your notice and suggestion of Reviewer #3, we mentioned the isolation site behind each isolate (Line 214-216, page 10).

15) From line 172 onwards - Replace the strain codes with their species names.

A15: Your recommendation was performed.

16) Lines 172-174: Firmicutes, Bacillaceae, Proteobacteria, Enterobacteriaceae, Actinobacteria, Actinomycetaceae are written in italics.

A16: The corrections were performed (Line 223-225, page 10).

C17) Lines 176-179 - The discussion is not strictly related to the results obtained.

A17: We referred to different bacterial groups (phylum Firmicutes, family Bacillaceae; phylum Proteobacteria, family Enterobacteriaceae and phylum Actinobacteria, family Actinomycetaceae) that related to the exact classification of our isolates and had the same scope (CaCO3 production with various applications). So, we thought the importance for mentioning other studies that are similar to our results. However, according to your notice, we stated such information in convenient way in the revised manuscript (line 227-229, page 10), particularly that there were different publications stated this point and dealt with their data with the same sort of discussion.

C18) Line 131 – please make clear the full composition of M9 media. Without such an information, it is impossible to verify the accuracy in the evaluation of precipitated CaCO3 (lines 238-239)

A18: Although we pointed out to the reference that mentioned the composition of M9 media, we followed your recommendation and reported the composition in the revised manuscript (Line 162-164, page 7-8). Additionally, the accuracy in the evaluation of precipitated CaCO3 (lines 238-239) was verified through abiotic control, without bacterial inoculum, (i.e. there was no chance for precipitation of media components even in absence of bacteria).

C19) Lines 189-190 – please consider the result obtained by Maciejewska et al. (2017) "Assessment of the Potential Role of Streptomyces in Cave Moonmilk Formation". Front. Microbiol. 8:1181. doi: 10.3389/fmicb.2017.01181

A19: We are grateful for your deep recommendation. We took the results of recommended publication by Maciejewska et al. (2017) in consideration. We would clarify that such study investigated all possible mechanisms that lead to CaCO3 precipitation in Moonmilk Cave. Where they studied ammonification of proteins, nitrate/ nitrite reduction, ureolysis and oxidative glucose breakdown but the actual application of such mechanisms in CaCO3 precipitation was only performed through ureolysis and protein ammonification. For more clarification, they detected only the previous five mechanisms to ensure their presence (without estimation of corresponding enzyme activity U/ml), but for CaCO3 precipitation, it was performed by ureolysis and protein ammonification only and not examined, characterized or monitored through nitrate reduction process, as we studied. Such might support our sentence in old manuscript. (Line 239-241, page 11).

20) From line 208 onward: results are presented and discussed in a very confusing manner preventing the comprehension of the text.

A20: Thanks for your comment. We took your valuable opinion in our consideration.

We revised and rewrote such part in simple, narrative and comprehensible way in the revised manuscript.

C21) The figures are so small and blurry that it is impossible to read them

A21: You are right. We apologize for such unintended mistake. Respect your opinion. All figures were adjusted and subjected to improve the image quality using photo editing software. All figures become clear enough to read and perceive.

C22) 237-244: the units of measure of the precipitated $CaCO_3$ need to conformed to an unique standard.

A22: Thanks for your critical observation. We followed your comment and made the required changes to confirm uniformity of unites according to standards (gm/ 100 mL) (Line 336-340, page 15).

C23) 247-252: the assumptions made by the authors seem to be of speculative nature. Are they any bibliographic references confirming their interpretation of the results?

A23: The lines 343-348 (Page 15, revised manuscript) described the obtained result and its proposed interpretation. As observed in Figure 3, there was increasing in the values of E.C, which could be attributed to the presence of ions such as $NO_2$-, $N_2O$-, $NO$-, $Ca_2$+ and $CO_3$- that were generated by the microbial activity and utilization of C/N substrates (sodium acetate and $Ca(NO_3)_2•4H_2O$). Generally, E.C of any solution increases with increasing of ions. In MICP studies, the ions in the media generated as a result of microbial activity on the substrates as indicated by several references. The references were added according to your recommendation (line 350, Page 16).

C24) 283-286: could the detected P derive from the ingredients used to make the culture broth?

A24: We thought that it isn't derived from media components, especially it presented only in vaterite samples and in considerable percentage, other than Na and Cl which were present in both vaterite and calcite samples in small percentage (0.5-0.66 %).

We recommended that it is biologically driven from bacterial cells which were incrusted by $CaCO_3$ stones. It represents essential constituent of bacterial biomolecules such as phospholipids, nucleic acids, proteins and/or polysaccharides. In addition, SEM images could confirm our suggestion. Where, the presence of calcified hyphae of Streptomyces and bacterial imprints of Raoultella planticola on vaterite spheres implied its biological nature.

C25) Please carefully check all the bibliographic references, that are not alway compliant with statement made by the authors.

A25: Thanks for your deep and thorough notice. All citations in the manuscript were revised in regards to their suitability and fitness to the mentioned interpretations and statements.

C 26) Wei et al. (2015), Wu et al (2011) and APHA, (1999) are not reported in the References section.

A26: The missed references were added.

C 27) Lines 241, 436 and 300 and reference section. Please replace "Kaur D.N." con "Dhami N.K."

A27: We followed your comment and all replacements were performed.

C28) All what was referred to Figures 3, 4, 5 and 6 are not verifiable. The Figures are small, blurry and Illegible.

A28: We apologize for this unintended mistake. The amendments and adjustments for such figures were performed in new revised manuscript.

C29) 297-298 and 303: how the authors can state that on Figure 6C and 6E a mucous matrix and mucous substance are evident?

A29: As refereed by arrows, there was slime like matrix or mucoid, curved string-like that link between calcified particles. Such mucoid strings were also calcified, so appeared thick. Microbiologically, it is known characteristic shape of exopolysaccharides that surround bacterial cells. Several literatures reported such shape:

• Dawson L., Valiente E., Faulds-Pain A., et al. 2012. Characterisation of Clostridium difficile Biofilm Formation, a Role for Spo0A. Plosone, 7, 12.

• J. Lam, r. Chan, k. Lam, and j. W. Costerton. 1980. Production of mucoid microcolonies by pseudomonas aeruginosa within infected lungs in cystic fibrosis. Infection and immunity, may 1980, 28, 2, 546-556.

• El Abed S., Ibnsouda S., Latrache H. and Hamadi F., 2012. Scanning Electron Microscopy (SEM) and Environmental SEM: Suitable Tools for Study of Adhesion Stage and Biofilm Formation. In: Scanning Electron Microscopy. V. Kazmiruk (ed). New York: INTECH Open Access Publisher.

• Wille, G., Hellal, J., Ollivier, P., Richard, A., Burel, A., Jolly, L., et al. (2017). Cryoscanning electron microscopy (SEM) and scanning transmission electron microscopy (STEM)-in-SEM for bio- and organo-mineral interface characterization in the environment. Microsc. Microanal. 23, 1159–1172.

• https://en.wikipedia.org/wiki/Biofilm

• https://phys.org/news/2018-09-key-bacterial-molecule-antibiotics.html

C30) 305-309: spores are notoriously inactive and cannot participate to Ca precipitation.

A30: In our study, it is probable that strain L. sphaericus entered into sporulation stage upon complete depletion of nutrients, which means that CaCO3 were formed by the vegetative cells before/during sporulation. As known, the vegetative cells transform to the spores under harsh conditions and return back again to vegetative upon removal of such conditions. Additionally, Jonkers et al., 2010 reported the application of Bacillus pseudofirmus DSM 8715 and B. cohnii DSM 6307 spores directly on the cement, proving that they remain viable for four months. Different literatures, listed below, reported

the applications of spores in soil solidification, bioplugging and biocementation, which were harsh and stress conditions.

• Jonkers, H. M., Thijssen, A., Muyzer, G., Copuroglu, O., & Schlangen, E. (2010). Application of bacteria as self-healing agent for the development of sustainable concrete. Ecol Eng, 36(2), 230–235.

• Anbu P., Kang C., Shin Y.and So J., 2016. Formations of calcium carbonate minerals by bacteria and its multiple applications.

• Ghosh T, Bhaduri S, Montemagno C, Kumar A. 2019. Sporosarcina pasteurii can form nanoscale crystals on cell surface. Plosone, 30, 14.

• Montano‑Salazar S., Lizarazo‑Marriaga J., Brandão P., 2017. Isolation and Potential Biocementation of Calcite Precipitation Inducing Bacteria from Colombian Buildings. Curr Microbiol.

C31) Lines 304-309. The statements do not make any sense. The spores in harsh condition remain spores and do nit evolve in vegetative forma, the only one able to contribute to the Ca precipitation.

A31: As we referred to the previous studies (direct above comment) dealing with the same issue, the spore suspension of different bacterial genera was applied in the harsh conditions and all of them proved the viability of the spores even after long time reached to months. So, we thought the importance of mentioning such sentence. Such characteristic property (sporulation) seemed to be advantageous, especially in the applications of soil stabilization and concrete healing (ongoing investigation in our lab).

Additionally, cells could be applied in the form of spores with its media and so could be transformed to vegetative cells that perform its function (e.g. CaCO3 deposition). Upon depletion of nutrients and entrance of relative harsh conditions (starvation)d, the cells don't loss their viability, but instead they transformed to spores till removal of such conditions.

C32) 313-316: the reference Hou et al. (2011) is not coherent with the discussion and it has been quoted in wrong way.

A32: From our point of view, this reference stated that the size of the precipitated calcite by Alternaria sp. was ranged from less than 1- and not exceeded 10 $\mu$m, which was agreed with the size of calcite crystals formed by strain Raoultella planticola (VIP), and under the same nitrate utilization conditions. Despite that, we followed your opinion and deleted it in the revised manuscript.

C33) 321: Rodriguez-Navarro et Al. (2012) obtained calcite and vaterite precipitation under experimental conditions widely differing from those described in this manuscript

A33: Totally agreed with your point of view. But, till our knowledge, there was no previous publication studied aerobic and anaerobic incubation's effect on CaCO3 deposition, in a comparative way, which we could take as a reference. So, we pointed out to the final results which was "variation in experimental conditions could result in variation in polymorph". Where, Rodriguez-Navarro et Al. (2012) mentioned that "at least in the systems studied, polymorph selection in bacterial calcium carbonate mineralization by heterotrophic bacteria is not bacterium or strain specific. Rather, under equal culture conditions, the nature of the substrate strongly influences which polymorph is formed".

We would manifest that there were recently published studies on aerobic and anaerobic MICP process, but the results indicated that calcite was only formed aerobically. While, the precipitation was neglected anaerobically so, its polymorph wasn't determined (Surabhi Jain & D. N. Arnepalli (2019); Lee et al., 2017). Besides, other recent studies just compare the kinetics of growth and precipitation between two conditions (Mitchell et al., 2019). Additionally, other studies reported formation of CaCO3 under only one condition of aeration (anaerobic) without testing it aerobically. • Surabhi Jain & D. N. Arnepalli (2019): Biochemically Induced Carbonate Precipitation in Aerobic and Anaerobic Environments by Sporosarcina pasteurii, Geomicrobiology Journal

• Yun Suk Lee, Hyun Jung Kim, and Woojun Park. Non-ureolytic calcium carbonate

precipitation by Lysinibacillus sp. YS11 isolated from the rhizosphere of Miscanthus sacchariflorus

• Andrew C. Mitchell1, Erika J. Espinosa-Ortiz, Stacy L. Parks, Adrienne J. Phillips, Alfred B. Cunninghamand Robin Gerlach. Kinetics of calcite precipitation by ureolytic bacteria under aerobic and anaerobic conditions. Biogeosciences, 16, 2147–2161, 2019.

• Bin Sun, Hui Zhao, Yanhong Zhao, Maurice E. Tucker, Zuozhen Han and Huaxiao Yan. 2020. Bio-Precipitation of Carbonate and Phosphate Minerals Induced by the Bacterium Citrobacter freundii ZW123 in an Anaerobic Environment. Minerals.

34) 327-328:please drop out. The sentence is useless in this context

A34: The sentence was deleted according to your recommendation

35) 335-483: the discussion is vague and superficial and carried out in a confusing way. In addition, the references quoted refer to article that used experimental protocols widely differing from those described in this manuscript .

A35: Thanks for your comment. We thoroughly revised the manuscript, reformated and reorganized it to avoid such critical points. Additionally, the references quoted were also revised.

36) Figure 6: The pictures are too small and difficult to read. Please consider a reduction of the photos by eliminating the redundant ones.

A36: Your observations were taken in consideration and adjusted in the revised manuscript to be readable and informative. Some pictures were deleted to reduce redundancy, while other contained some informative details.